# Higher-genus Fay-like identities from meromorphic generating functions

Konstantin Baune, Johannes Broedel, Egor Im,
Artyom Lisitsyn, Yannis Moeckli

*Institute for Theoretical Physics, ETH Zurich*
*Wolfgang-Pauli-Str. 27, 8093 Zurich, Switzerland*

`baunek@ethz.ch, jbroedel@ethz.ch, egorim@ethz.ch,`
`alisitsyn@ucdavis.edu, moeckliy@ethz.ch`

## Abstract

A possible way of constructing polylogarithms on Riemann surfaces of higher genera facilitates integration kernels, which can be derived from generating functions incorporating the geometry of the surface. Functional relations between polylogarithms rely on identities for those integration kernels.

In this article, we derive identities for Enriquez' meromorphic generating function and investigate the implications for the associated integration kernels. The resulting identities are shown to be exhaustive and therefore reproduce all identities for Enriquez' kernels conjectured in `arXiv:2407.11476` recently.

# 1  Introduction

Expressing observables in quantum field theories and string theories in an efficient way requires the use of polylogarithms on Riemann surfaces of arbitrary genus [1–7]: these functions allow linking parameters of the observables to geometric information determining the Riemann surface in question. Simultaneously, polylogarithms offer the possibility to mathematically implement the inevitable branch cut structure appearing in observables when considered as functions of – for example – Mandelstam variables and masses of particles.

One possible and viable way to construct polylogarithms is as iterated integrals [8] over a class of integration kernels appropriate for the Riemann surface in question. Usually, this class of kernels can be derived from a generating function, which in turn gives rise to a connection on the Riemann surface. Flatness of this connection form is sufficient for homotopy invariance of the associated polylogarithms.

In most situations the generating function does not only satisfy flatness of the associated connection form, but is subject to further relations and identities. At genus one, those are the so-called Fay identities for the Kronecker function [9, Prop. 5 (iii)], which can be derived from the Fay trisecant equations [10, 11]. For genera higher than one, the connection between the Fay trisecant equation and the identities explored in this article remains unclear. The anticipation of establishing such a link later led us to call the kernel identities derived from higher-genus generating functions *Fay-like*.

For each generating function and associated connection, the resulting class of polylogarithms needs to be proven to be complete, that is, it shall be closed under integration and differentiation.

Suitable connection forms on Riemann surfaces of genus zero and genus one have been identified and explored in [12] and [13–16]. Taking these connection forms as a starting point, a vast literature is available for the construction of polylogarithms on Riemann surfaces of genus zero and genus one, see for example [17–19] and [20–23, 9], respectively. For the construction of connection forms and associated polylogarithms on Riemann surfaces of genus greater than one, several approaches and suggestions have appeared in the last years [15, 16, 24–31].

The connection forms underlying the constructions of polylogarithms live on a principal bundle of the Riemann surface and therefore take values in an associative algebra, whose generators are associated to the non-trivial cycles of the Riemann surface. While the subalgebra associated to the $\mathfrak{B}$-cycles (which is relevant to the quasi-periodicities of the connection) is abelian at genus one, it will become non-abelian at higher genera, because generators associated to different cycles with non-trivial quasi-periodicity do not commute. At all genera, the choice of generators for the algebra is not unique and correspondingly, there are several representations for generating functions. In addition, there is as well choices for the analytic properties of the generating functions, which implies the analytic properties of the associated integration kernels: integration kernels can either be *meromorphic* or *periodic* in all of the cycles of the Riemann surface. Furthermore one can choose the integration kernels to exhibit simple poles only or allow for higher poles.

In summary, knowing a generating function that incorporates the geometry of the Riemann surface and gives rise to a flat connection form, allows for the construction of polylogarithms on this particular Riemann surface. The properties of the connection form (including its flatness) will then imply relations between integration kernels, which in turn lead to functional relations between polylogarithms and associated special values. Proving closure of the polylogarithmic space thus constructed leads to a class of polylogarithms on a Riemann surface.

An important tool when exploring classes of polylogarithms is functional relations between

them. In a physics context, functional relations allow for canonical (and thus comparable) representation of observables. In retrospective, the largest structural advantage when constructing polylogarithms as iterated integrals is the algebraization: the associated algebraic structures lead to an algorithmic way of using functional identities between polylogarithms to reduce them to a canonical set. Those algorithms have been described for polylogarithms at genus zero [32,33] and genus one [34–37]. Some of those relations have been used in the (easier) context of exploring relations between various classes of (multiple) zeta values, see for example [38–42].

Functional relations between polylogarithms based on relations between integration kernels have been put to use in numerous contexts in physics, in particular in the calculation of string scattering amplitudes as well as for numerous calculations of Feynman integrals in quantum field theory (see the review articles [43] and [44], respectively).

For periodic and single-valued kernels, a set of Fay-like identities has been proven to be valid in ref. [45] recently. In the same reference, so-called interchange lemmas were proven and Fay-like identities have been conjectured for meromorphic versions of kernels. In this article, we are going to investigate the identities for meromorphic versions of integration kernels using the language of generating functions based on Enriquez' connection [26]. Employing our formalism at genus one, we recover the known elliptic Fay identities. Upon expansion into the meromorphic kernels suggested by Enriquez, our identities are shown to comprise all identities put forward in ref. [45]. We are going to derive and prove several classes of identities and show that there cannot be any identities other than those we have been considering.

The structure of this article is as follows: starting from Enriquez' generating function, which is reviewed in § 2, we are defining so-called *component forms*, which are substructures of the original generating function, and allow incorporation of a second algebra for the quasi-periodicities. The definition and properties of these component forms, as well as their Hopf algebra structure is content of § 3, together with a couple of basic identities and algebraic contractions of the component forms.

The main results of our article are contained in § 4, which are identities for the component forms. The principal statements are collected in Theorems 9 and 12 along with the respective proofs. Furthermore, § 5 is aimed at exploring identities for integration kernels derived from the identities for component forms. We combine our identities to reproduce the identities suggested in ref. [45]. In § 6 we connect and interpret the language of the current article to the well-explored formalism for building polylogarithms at genus one. Based on identities for Enriquez' integration kernels, we show examples of functional relations for higher-genus polylogarithms in § 7. Finally, we summarize and formulate various open questions in § 8. Several details of long manipulations are relegated from the respective proofs to various appendices.

## 2   Enriquez' connection

We consider a Riemann surface $\Sigma$ of genus $h$ with homology basis $\mathfrak{A}_i$, $\mathfrak{B}_i$, for $i \in \{1 \dots h\}$, and associated differentials $\omega_i$ such that

$$\oint_{\mathfrak{A}_i} \omega_j = \delta_{ij}, \qquad \oint_{\mathfrak{B}_i} \omega_j = \tau_{ij}. \tag{2.1}$$

All objects defined below depend on the geometry of the Riemann surface in question. The precise way how this geometric dependence is captured depends on the choice of assigning quasi-periodicities to the cycles in the homology basis. Within this article we are going to choose

the $\mathfrak{A}$-cycles to have trivial quasi-periodicities, where the $\mathfrak{B}$-cycles will carry the non-trivial geometric information through their quasi-periodicities. This trivialization of the $\mathfrak{A}$-cycles is a crucial ingredient when constructing the Schottky uniformization of Riemann surfaces [46]. Below, we are going to formally assign algebra generators $a_i$ and $b_i$ to the quasi-periodicities when walking around $\mathfrak{A}$-cycles and $\mathfrak{B}$-cycles, respectively. Usually only the non-trivial behavior when going around $\mathfrak{B}$-cycles is noted.

Let us consider the free associative algebra $\mathfrak{t}$ generated by the set $\{a_i, b_i\}_{i=1}^{h}$. This algebra reflects the structure of the fundamental group of a punctured Riemann surface $\Sigma \setminus \{x\}$, where $x \in \Sigma$ denotes the puncture. In the following, we consider a unique one-form valued in the algebra $\mathfrak{t}$, which will serve as a generating function for the integration kernels used to define higher-genus polylogarithms later. In what follows, we will mostly consider the subalgebra $\mathfrak{b} \subset \mathfrak{t}$ generated by $\{b_i\}_{i=1}^{h}$ only.

**Definition 1** (Enriquez–Zerbini [27, Prop. 3.13]). *There exists a unique meromorphic one-form $K(z, x)$ in $z$, valued in the algebra $\mathfrak{t}$, for $z, x$ in $\Sigma^{\mathrm{univ}}$, the universal cover of $\Sigma$, defined by the two properties:*

$$K(\gamma_i z, x) = e^{b_i} K(z, x), \quad \forall i \in \{1 \dots h\}, \tag{2.2a}$$

$$(-2\pi \mathring{\imath}) \operatorname*{Res}_{z=x} K(z, x) = \sum_{j=1}^{h} b_j a_j, \tag{2.2b}$$

*where $\gamma_i z$ is the image of $z$ under moving around the cycle[1] $\mathfrak{B}_i$. We are going to refer to $K(z, x)$ as Enriquez' connection[2].*

This defines a holomorphic connection on a punctured Riemann surface $\Sigma \setminus \{x\}$ with fundamental group freely generated by the cycles $\{\mathfrak{A}_i, \mathfrak{B}_i\}_{i=1}^{h}$. Restricting to genus one, the above connection valued in $\mathfrak{t}$ is related to the Kronecker function appearing in ref. [5, 14] and used in the construction of a single-valued connection in ref. [9].

Enriquez' connection $K(z, x)$ can be expanded into words composed from the algebra generators of $\mathfrak{t}$. The corresponding coefficients $\omega_{i_1 \cdots i_r j}(z, x)$ are one-forms in $z$ and functions in $x$ and are referred to as *integration kernels*,

$$\begin{aligned} K(z, x) &= \sum_{r=0}^{\infty} \sum_{i_1, \cdots, i_r, j=1}^{h} \omega_{i_1 \cdots i_r j}(z, x) \, b_{i_1} \cdots b_{i_r} a_j \\ &= \sum_{r=0}^{\infty} \omega_{i_1 \cdots i_r j}(z, x) \, b_{i_1} \cdots b_{i_r} a_j. \end{aligned} \tag{2.3}$$

Here and in the remainder of the article repeated indices are summed over from 1 to $h$ implicitly unless otherwise noted. This is as well understood, if the two indices belong to the same object (e.g. in $\omega_{ijj}$).

In ref. [26], the integration kernels are shown to satisfy the following quasi-periodicity prop-

---

[1]The notation for $\gamma_i z = z + \mathfrak{B}_i$ is borrowed from ref. [31], where on the Schottky cover going around the cycle $\mathfrak{B}_i$ is equivalent to applying a Schottky group generator $\gamma_i$.

[2]In the original article ref. [26] this connection is defined on a Riemann surface without punctures, which has a different fundamental group and, hence, gets an additional constraint on the algebra $\mathfrak{t}$, see ref. [31] for more details.

erties:

$$\omega_{i_1\cdots i_r j}(\gamma_k z, x) = \sum_{l=0}^{r} \frac{1}{l!} \delta_{ki_1\cdots i_l} \, \omega_{i_{l+1}\cdots i_r j}(z, x), \tag{2.4a}$$

$$\omega_{i_1\cdots i_r ij}(z, \gamma_k x) = \omega_{i_1\cdots i_r ij}(z, x) + \delta_{ij} \sum_{l=0}^{r} \frac{(-1)^{l+1}}{(l+1)!} \delta_{ki_r\cdots i_{r-l+1}} \, \omega_{i_1\cdots i_{r-l}k}(z, x), \tag{2.4b}$$

where $\delta_{ki_1\cdots i_l} = \prod_{n=1}^{l} \delta_{ki_n}$. Furthermore, the kernels satisfy the residue condition

$$(-2\pi\mathring{\imath}) \operatorname*{Res}_{z=x} \omega_{i_1\cdots i_r ij}(z, x) = \delta_{r0}\delta_{ij}, \tag{2.5}$$

which implies that poles are carried by two-index kernels with equal indices exclusively. Considering the behavior of kernels under cyclic shifts, it becomes clear that poles on the r.h.s. of eqs. (2.4a) and (2.4b) propagate to the l.h.s. at positions related to the original $z$ and $x$ by moving around cycles. Accordingly, eq. (2.5) takes into account only the pole in the fundamental domain.

In general, higher-genus polylogarithms can be built recursively as iterated integrals from integration kernels [26–29]. In the notation of ref. [31], polylogarithms are built from the kernels in eq. (2.3) via

$$\tilde{\Gamma} \left( \begin{smallmatrix} \mathbf{i}_{n_1}, \ldots, \mathbf{i}_{n_k} \\ x_1, \ldots, x_k \end{smallmatrix} ; z \right) = \int_{t=z_0}^{z} \omega_{\mathbf{i}_{n_1}}(t, x_1) \, \tilde{\Gamma} \left( \begin{smallmatrix} \mathbf{i}_{n_2}, \ldots, \mathbf{i}_{n_k} \\ x_2, \ldots, x_k \end{smallmatrix} ; t \right), \qquad \tilde{\Gamma} \left( \; ; z \right) = 1, \tag{2.6}$$

where $\mathbf{i}_{n_\ell}$ are the multi-indices of the integration kernels $\omega$ of length $n_\ell$. The points $x_\ell$ in the second line denote the second parameter of these kernels. The point $z_0 \in \Sigma^{\mathrm{univ}}$ is the fixed basepoint of integration.

For the remainder of this article, we will be concerned with relations among various generating functions for integration kernels. The simplest set of such relations, leading to linear 3-point identities for integration kernels, will be reviewed in the next subsection.

## 2.1 Linear 3-point identities

Before studying Fay-like quadratic identities in § 4, it is instructive to consider the space of linear 3-point identities satisfied by the integration kernels $\omega_{i_1\cdots i_r j}(z, x)$.

Several of these identities have already been shown in ref. [26, Lemma 9], where one finds[3]

$$\omega_{i_1\cdots i_r jk}(z, x) - \omega_{i_1\cdots i_r jk}(z, y) = 0, \quad \text{for } j \neq k, \tag{2.7a}$$

$$\left( \left( \omega_{i_1\cdots i_r pp'}(z, x) - \omega_{i_1\cdots i_r qq'}(z, x) \right) - \left( \omega_{i_1\cdots i_r pp'}(z, y) - \omega_{i_1\cdots i_r qq'}(z, y) \right) \right) \Big|_{\substack{p'=p \\ q'=q}} = 0. \tag{2.7b}$$

The above equations show that holomorphic (combinations of) kernels are independent of the location of the pole. Accordingly, we will omit the second argument for holomorphic kernels in the following. As an immediate application, let us show that any linear 3-point identity can be traced back to the pole-independence conditions above.

**Proposition 2** (Triviality of linear identities). *Any linear 3-point identity between integration kernels can be written as a linear combination of identities eq. (2.7a) and eq. (2.7b).*

---

[3]The notation in eq. (2.7b) is our convention to write repeated indices which are not summed over.

*Proof.* For the proof, define a linear 3-point identity $\mathcal{L}^{(m)}(z, y, x)$ of weight $m$ as

$$\mathcal{L}^{(m)}(z, y, x) = \sum_{r=0}^{m} \lambda_{i_1 \ldots i_r j}\, \omega_{i_1 \ldots i_r j}(z, x) + \eta_{i_1 \ldots i_r j}\, \omega_{i_1 \ldots i_r j}(z, y) = 0, \tag{2.8}$$

whose vanishing is ensured by a particular fixed choice of coefficients $\lambda_{i_1 \ldots i_r j}, \eta_{i_1 \ldots i_r j} \in \mathbb{C}$, where repeated indices are summed over implicitly. In the following we will prove validity of the theorem inductively over the weight $m$.

Validity of the induction start $m = 0$ is evident as the holomorphic differentials are linearly independent, which implies all coefficients in any weight-zero identity to vanish.

For the inductive step, we assume that for $\mathcal{L}^{(\ell)}(z, y, x) = 0$, for all $0 \leq \ell < m$, the theorem holds. Let us define recursively a collection of lower weight identities[4]

$$\mathcal{L}_{i_1 \cdots i_k}^{(m-k)} = \mathcal{L}_{i_1 \cdots i_{k-1}}^{(m-k+1)}(\gamma_{i_k} z, y, x) - \mathcal{L}_{i_1 \cdots i_{k-1}}^{(m-k+1)}(z, y, x) = 0, \quad \text{for } k = 1, \ldots, m, \tag{2.9}$$

each of which are true for arbitrary choices of $i_1 \cdots i_k$. With each step, the quasi-periodicity eq. (2.4a) produces extra terms of lower weight, and the highest-weight terms are canceled out, reducing the weight of the identity by one. Isolating the highest-weight terms, we can write

$$\mathcal{L}_{i_1 \cdots i_k}^{(m-k)} = \lambda_{i_1 \ldots i_m j}\, \omega_{i_{k+1} \ldots i_m j}(z, x) + \eta_{i_1 \ldots i_m j}\, \omega_{i_{k+1} \ldots i_m j}(z, y)$$
$$+ \text{[lower-weight terms]} = 0. \tag{2.10}$$

Choosing $k = m$ yields a collection of weight-zero identities, for which linear independence of the holomorphic forms $\omega_j(z)$ implies the following conditions on the coefficients $\lambda$ and $\eta$:

$$\mathcal{L}_{i_1 \cdots i_m}^{(0)} = (\lambda_{i_1 \ldots i_m j} + \eta_{i_1 \ldots i_m j})\, \omega_j(z) = 0 \implies \lambda_{i_1 \ldots i_m j} + \eta_{i_1 \ldots i_m j} = 0. \tag{2.11}$$

Taking the residue of the collection of weight-one identities obtained by choosing $k = m - 1$, one deduces a second set of conditions on the coefficients $\lambda$ and $\eta$,

$$\operatorname*{Res}_{z=x} \mathcal{L}_{i_1 \cdots i_{m-1}}^{(1)} = \operatorname*{Res}_{z=x} \left( \lambda_{i_1 \ldots i_m j}\, \omega_{i_m j}(z, x) + \eta_{i_1 \ldots i_m j}\, \omega_{i_m j}(z, y) + \text{[weight-zero terms]} \right)$$
$$= -\frac{1}{2\pi i}\, \lambda_{i_1 \ldots i_{m-1} jj} = 0 \implies \sum_{j=2}^{h} \lambda_{i_1 \ldots i_{m-1} jj} = -\lambda_{i_1 \ldots i_{m-1} 11}, \tag{2.12}$$

where only the first term contributes to the residue as all the other terms are regular at $z = x$.

Using the conditions on $\lambda$ and $\eta$ in eqs. (2.11) and (2.12) for the highest weight, one can rewrite $\mathcal{L}^{(m)}$ from eq. (2.8) as

$$\mathcal{L}^{(m)}(z, y, x)$$
$$= \sum_{j \neq i_m} \lambda_{i_1 \ldots i_m j} \left( \omega_{i_1 \ldots i_m j}(z, x) - \omega_{i_1 \ldots i_m j}(z, y) \right)$$
$$+ \sum_{j=2}^{h} \lambda_{i_1 \ldots i_{m-1} jj} \left( [\omega_{i_1 \ldots i_m jj}(z, x) - \omega_{i_1 \ldots i_m 11}(z, x)] - [\omega_{i_1 \ldots i_m jj}(z, y) - \omega_{i_1 \ldots i_m 11}(z, y)] \right)$$
$$+ \mathcal{L}^{(m-1)}(z, y, x), \tag{2.13}$$

where one can observe the weight-$m$ terms to vanish due to eqs. (2.7a) and (2.7b). Accordingly, combining the translational behavior of the integration kernels and validity of the weight-$m$ identity $\mathcal{L}^{(m)}(z, y, x) = 0$ gives rise to an identity

$$\mathcal{L}^{(m-1)}(z, y, x) = \sum_{r=0}^{m-1} \lambda_{i_1 \ldots i_r j}\, \omega_{i_1 \ldots i_r j}(z, x) + \eta_{i_1 \ldots i_r j}\, \omega_{i_1 \ldots i_r j}(z, y) = 0 \tag{2.14}$$

of weight $m-1$, with the particular fixed choice of coefficients $\lambda_{i_1 \cdots i_r j}, \eta_{i_1 \cdots i_r j}$ assumed for $\mathcal{L}^{(m)}$ in eq. (2.8).

By the assumption of the induction step, the theorem holds for $\mathcal{L}^{(m-1)}(z, y, x)$. Therefore, one concludes that $\mathcal{L}^{(m)}(z, y, x)$ can be written as a linear combination of eqs. (2.7a) and (2.7b) for any weight $m \geq 0$. $\qquad \square$

Notice that the above proof also shows that any linear identity is graded by weight, that is, in eq. (2.8) each contribution for a particular weight $r \in \mathbb{N}$ vanishes individually:

$$\lambda_{i_1 \cdots i_r j}\, \omega_{i_1 \cdots i_r j}(z, x) + \eta_{i_1 \cdots i_r j}\, \omega_{i_1 \cdots i_r j}(z, y) = 0\,. \tag{2.15}$$

This is due to the fact that the above theorem is proven inductively modulo lower-weight terms relying on eqs. (2.7a) and (2.7b) of particular weight $r$.

Validity of Proposition 2 has been shown using identities (2.7) for individual kernels. These identities will be rewritten in terms of generating functions in eqs. (3.19) and (3.20).

# 3 Component forms

In order to show quadratic relations between generating functions for Enriquez' integration kernels, we will define so-called *component forms*, which are generating functions for specific combinations of integration kernels, where certain indices are fixed. This allows for expressions involving contractions of free indices. However, before doing so we briefly introduce the Hopf algebra structure on the free associative algebra $\mathfrak{b}$ that will prove to be a useful tool in what follows.

## 3.1 Hopf algebra

Component forms are best described as one-forms valued in the tensor algebra of $\mathfrak{b}$. The algebra $\mathfrak{b}$ has a natural Hopf algebra structure, which allows for systematic investigation of the component forms. The Hopf algebraic structure of $\mathfrak{b}$ was already exploited in ref. [27].

**Definition 3.** *A Hopf algebra[5] $\mathfrak{h}$ (over a field $\mathbb{K}$) is a bialgebra together with a (unique) anti-homomorphism map $S : \mathfrak{h} \to \mathfrak{h}$ called the* antipode *such that the algebra structure (*product $\mu : \mathfrak{h} \otimes \mathfrak{h} \to \mathfrak{h}$ and unit $\eta : \mathbb{K} \to \mathfrak{h}$*), coalgebra structure (*coproduct $\Delta : \mathfrak{h} \to \mathfrak{h} \otimes \mathfrak{h}$ and counit $\epsilon : \mathfrak{h} \to \mathbb{K}$*) and the antipode are all compatible. This means that the coproduct $\mu$ and counit $\epsilon$ must be algebra homomorphisms and the antipode $S$ satisfies*

$$\mu \circ (S \otimes \mathrm{id}) \circ \Delta = \mu \circ (\mathrm{id} \otimes S) \circ \Delta = \eta \circ \epsilon. \tag{3.1}$$

For the free associative algebra $\mathfrak{b}$ defined at the beginning of §2 and generated by elements $b_i$ with $i \in \{1, \ldots, h\}$, the canonical Hopf algebra structure has the standard product and unit maps. The coproduct, counit and antipode acting on the generators of $\mathfrak{b}$ and the neutral element are defined as

$$\Delta(b_i) = b_i \otimes 1 + 1 \otimes b_i, \quad \Delta(1) = 1 \otimes 1, \quad \epsilon(b_i) = 0, \quad \epsilon(1) = 1, \quad S(b_i) = -b_i, \quad S(1) = 1. \tag{3.2}$$

These relations are extended to the whole of $\mathfrak{b}$ by linearity and (anti-)homomorphism property.

---

[4]Notice that despite using the same letter, objects $\mathcal{L}$ with and without subindices are different.
[5]See for example ref. [47] for a general introduction.

In the remainder of the article we will use the coproduct and antipode to construct maps for generating functions in order to derive several useful formulas and identities.

## 3.2 First component form

**Definition 4.** *Following the notation in refs. [27, 31], the $j$-th component $K_{(\mathfrak{b})j}$ of Enriquez' connection, which is referred to as the* first component form, *valued in the free associative algebra $\mathfrak{b} \subset \mathfrak{t}$, is defined as*[6]

$$K_{(\mathfrak{b})j}(z,x) = \sum_{r \geq 0} \omega_{i_1 \cdots i_r j}(z,x)\, b_{i_1} \cdots b_{i_r}\,, \quad j \in \{1, \ldots, h\}\,, \tag{3.3}$$

*where as before $\{b_i\}_{i=1}^h$ are the generators of the algebra $\mathfrak{b}$.*

Comparing with eq. (2.3), it is clear that Enriquez' connection is a contraction of the first component forms $K_{(\mathfrak{b})j}$ with the letters $a_j$,

$$K(z,x) = K_{(\mathfrak{b})j} a_j. \tag{3.4}$$

Using quasi-periodicity (2.4) and residue (2.5) of the integration kernels within eq. (3.3), one can derive the quasi-periodicity and residue of the first component form to be

$$K_{(\mathfrak{b})j}(\gamma_k z, x) = e^{b_k} K_{(\mathfrak{b})j}(z,x), \tag{3.5a}$$

$$K_{(\mathfrak{b})j}(z, \gamma_k x) = K_{(\mathfrak{b})j}(z,x) + K_{(\mathfrak{b})k}(z,x)\, \frac{e^{-b_k} - 1}{b_k}\, b_j, \tag{3.5b}$$

$$(-2\pi i)\operatorname*{Res}_{z=x} K_{(\mathfrak{b})j}(z,x) = b_j, \tag{3.5c}$$

where there is no summation over $k$ in eq. (3.5b) and $\gamma_k$ again denotes the operation of moving along the cycle $\mathfrak{B}_k$. The details of the calculation are spelled out in Appendix A.1.

As a side note, upon acting with the counit on the first component form, one obtains the holomorphic differential

$$\epsilon(K_{(\mathfrak{b})j}(z,x)) = \omega_j(z)\,. \tag{3.6}$$

## 3.3 Second component form

**Definition 5.** *We define the* second component form $K_{(\mathfrak{b}_1)j(\mathfrak{b}_2)k}$, *valued in the tensor algebra $\mathfrak{b} \otimes \mathfrak{b} = \mathfrak{b}^{\otimes 2}$, as*

$$K_{(\mathfrak{b}_1)j(\mathfrak{b}_2)k}(z,x) = \sum_{r,s \geq 0} \omega_{i_1 \cdots i_r j p_1 \cdots p_s k}(z,x)\, b_{i_1} \cdots b_{i_r} \otimes b_{p_1} \cdots b_{p_s}, \quad j,k \in \{1,\ldots,h\}. \tag{3.7}$$

The notation $(\mathfrak{b}_1)$ and $(\mathfrak{b}_2)$ in eq. (3.7) explains the position of the corresponding algebra generators within the tensor product[7]. In what follows we will enumerate the algebra label in the first component form too, whenever the position of the generators in a tensor product is not obvious.

Similar to the first component form above, we can calculate the quasi-periodicity and the

---

[6]This notion of first component form has also been considered in ref. [27].
[7]The total number of tensor sites is understood from the context.

residue of the second component form. We find that (cf. Appendix A.1)

$$K_{(\mathfrak{b}_1)j(\mathfrak{b}_2)k}\left(\gamma_i z, x\right) = e^{b_i} K_{(\mathfrak{b}_1)j(\mathfrak{b}_2)k}(z,x) + \delta_{ij} \frac{1 \otimes e^{b_i} - e^{b_i} \otimes 1}{1 \otimes b_i - b_i \otimes 1} K_{(\mathfrak{b}_2)k}(z,x), \qquad (3.8)$$

where there is no summation over $i$ and

$$(-2\pi \mathring{i}) \operatorname*{Res}_{z=x} K_{(\mathfrak{b}_1)j(\mathfrak{b}_2)k}(z,x) = \delta_{jk}(1 \otimes 1). \qquad (3.9)$$

A special case of the second component form is obtained when it is acted upon with the counit on the second tensor site

$$K_{(\mathfrak{b})jk}(z,x) := (\mathrm{id} \otimes \epsilon)\left(K_{(\mathfrak{b}_1)j(\mathfrak{b}_2)k}(z,x)\right) = \epsilon_2\left(K_{(\mathfrak{b}_1)j(\mathfrak{b}_2)k}(z,x)\right) = \sum_{r \geq 0} \omega_{i_1 \cdots i_r jk}(z,x)\, b_{i_1} \cdots b_{i_r},$$
$$(3.10)$$

where we used the notation for a map $m$ that $m_{p_1 \ldots p_n}$ acts non-trivially only on the sites $p_1, \ldots, p_n$. Contraction of the special form in eq. (3.10) over $\mathfrak{b}$ yields the first component form:

$$K_{(\mathfrak{b})k}(z,x) = K_{(\mathfrak{b})jk}(z,x)b_j. \qquad (3.11)$$

Quasiperiodicity $K_{(\mathfrak{b})jk}(\gamma_i z, x)$ for the special form can be easily deduced from eq. (3.8). The expression for the quasi-periodicity of the full second component form (3.7), $K_{(\mathfrak{b}_1)j(\mathfrak{b}_2)k}(z, \gamma_i x)$, is involved and will not be used in the rest of the article. Instead, we note quasi-periodicity in the auxiliary variable for the special form

$$K_{(\mathfrak{b})jk}\left(z, \gamma_i x\right) = K_{(\mathfrak{b})jk}(z,x) + \delta_{jk} K_{(\mathfrak{b})i}(z,x) \frac{e^{-b_i} - 1}{b_i}, \qquad (3.12)$$

where again there is no summation over $i$. A detailed derivation of the properties in eqs. (3.8), (3.9) and (3.12) is noted in Appendix A.1.

Note, that in principle one could define a $k$-th component form valued in $\mathfrak{b}^{\otimes k}$. However, we are not going to consider this possibility here, as the second component form is sufficient for deriving the Fay-like identities in this article.

## 3.4 Hopf algebra operations on component forms

This subsection is dedicated to understanding the additional structures and properties of the component forms arising from applying various Hopf algebra maps. This will allow us to construct variations of the component forms, build relations between them and map between different identities for generating functions.

**Antipode.** One of the tools necessary for the identities in § 4 will be the antipode map. Since it is an anti-homomorphism, application of the antipode on a component form effectively reverses the order of the indices of the integration kernels in the expansion. For the first and second component forms one finds

$$K_{(S\mathfrak{b})j}(z,x) := S\left(K_{(\mathfrak{b})j}(z,x)\right) = \sum_{r \geq 0} (-1)^r \omega_{i_r \cdots i_1}\, b_{i_1} \cdots b_{i_r}, \qquad (3.13a)$$

$$K_{(S\mathfrak{b}_1)j(\mathfrak{b}_2)k}(z,x) := S_1\left(K_{(\mathfrak{b}_1)j(\mathfrak{b}_2)k}(z,x)\right) = \sum_{r,s \geq 0} (-1)^r \omega_{i_r \cdots i_1 j p_1 \cdots p_s k}\, b_{i_1} \cdots b_{i_r} \otimes b_{p_1} \cdots b_{p_s},$$
$$(3.13b)$$

$$K_{(\mathfrak{b}_1)j(S\mathfrak{b}_2)k}(z,x) := S_2\left(K_{(\mathfrak{b}_1)j(\mathfrak{b}_2)k}(z,x)\right) = \sum_{r,s\geq 0}(-1)^s\omega_{i_1\cdots i_r j p_s\cdots p_1 k}\, b_{i_1}\cdots b_{i_r}\otimes b_{p_1}\cdots b_{p_s}.$$

$$(3.13c)$$

**Coproduct.** Another useful set of results comes from applying the coproduct on the component forms. By explicitly evaluating the coproduct of the first component form eq. (3.3), one finds

$$
\begin{aligned}
K_{(\Delta\mathfrak{b}_{12})j}(z,x) :=&\, \Delta\left(K_{(\mathfrak{b})j}(z,x)\right) = \sum_{r\geq 0}\omega_{i_1\cdots i_r j}(z,x)\,\Delta(b_{i_1})\cdots\Delta(b_{i_r})\\
=&\, \sum_{r,s\geq 0}\omega_{(i_1\cdots i_r \shuffle p_1\cdots p_s)j}(z,x)\, b_{i_1}\cdots b_{i_r}\otimes b_{p_1}\cdots b_{p_s}\,,
\end{aligned}
$$

$$(3.14)$$

where $\shuffle$ denotes the shuffle product[8]. Here we used the subscript for the label $(\Delta\mathfrak{b})$ in order to denote the position of the generators in a tensor product *after* application of the coproduct. Acting with the coproduct $\Delta$ on either of the tensor sites in eq. (3.7) yields

$$
\begin{aligned}
K_{(\Delta\mathfrak{b}_{12})j(\mathfrak{b}_3)k}(z,x) :=&\, \Delta_1\left(K_{(\mathfrak{b}_1)j(\mathfrak{b}_2)k}(z,x)\right)\\
=&\, \sum_{r,s,l\geq 0}\omega_{(i_1\cdots i_r \shuffle j_1\cdots j_s)j p_1\cdots p_l k}\, b_{i_1}\cdots b_{i_r}\otimes b_{j_1}\cdots b_{j_s}\otimes b_{p_1}\cdots b_{p_l}, &(3.15a)\\
K_{(\mathfrak{b}_1)j(\Delta\mathfrak{b}_{23})k}(z,x) :=&\, \Delta_2\left(K_{(\mathfrak{b}_1)j(\mathfrak{b}_2)k}(z,x)\right)\\
=&\, \sum_{r,s,l\geq 0}\omega_{i_1\cdots i_r j(j_1\cdots j_s \shuffle p_1\cdots p_l)k}\, b_{i_1}\cdots b_{i_r}\otimes b_{j_1}\cdots b_{j_s}\otimes b_{p_1}\cdots b_{p_l}. &(3.15b)
\end{aligned}
$$

The resulting forms live in the triple tensor product $\mathfrak{b}^{\otimes 3}$. When exploring quadratic identities in §4.2 below, we will have to consider the double product $\mathfrak{b}^{\otimes 2}$ only, to which the above forms can be mapped to by applying the product map $\mu$ to any two of the three tensor sites.

Quasi-periodicities and residues for the above objects can be deduced from linearity of the coproduct. An example containing a similar calculation is shown in Appendix A.1.

**Product and permutation.** Application of the permutation operator $\mathcal{P}_{pq}$ and the product operator $\mu_{pq}$ is evident: the former permutes tensor sites $p$ and $q$, while the latter concatenates the generators at sites $p$ and $q$ (only for neighboring sites $q = p+1$). The notation with enumerated algebra labels $(\mathfrak{b}_i)$ allows to define

$$
\begin{aligned}
K_{(\mathfrak{b}_2)j(\mathfrak{b}_1)k} &:= \mathcal{P}_{12}\, K_{(\mathfrak{b}_1)j(\mathfrak{b}_2)k},\\
K_{(\mathfrak{b}_1)j(\mathfrak{b}_1)k} &:= \mu_{12} K_{(\mathfrak{b}_1)j(\mathfrak{b}_2)k}.
\end{aligned}
$$

$$(3.16)$$

Notice a possible ambiguity: despite yielding different results, the action of $\mu_{12}$ and $\mu_{12}\circ\mathcal{P}_{12}$ cannot be distinguished with this notation. Throughout this article the concatenation of algebra generators will be assumed to be in the order, in which the corresponding algebra labels appear.

**Contractions of component forms.** We have already seen in eq. (3.11) that the second component form is related to the first component form upon acting with the counit and contracting the first free index $j$ with a corresponding algebra generator $b_j$. Combining the coproduct map together with contractions gives rise to more interesting results, summarized in the following proposition.

---

[8]The shuffle of two ordered sets $P$ and $Q$ is obtained by selecting from $\mathrm{Perm}(P\cap Q)$ those permutations, which keep the individual orderings of the sets $P$ and $Q$ intact. If $P$ and $Q$ are index sets, a summation over the objects indexed by the shuffles is implied.

**Proposition 6.** *The following contraction identities for the first and second component forms hold:* [9]

$$K_{(\Delta\mathfrak{b}_{12})k}(z,x) - K_{(\mathfrak{b}_1)k}(z,x) = K_{(\Delta\mathfrak{b}_{12})j(\mathfrak{b}_1)k}(z,x)(1 \otimes b_j), \tag{3.17a}$$

$$K_{(\Delta\mathfrak{b}_{12})k}(z,x) - K_{(\mathfrak{b}_1)k}(z,x) = (1 \otimes b_j)K_{(\mathfrak{b}_1)j(\Delta\mathfrak{b}_{12})k}(z,x), \tag{3.17b}$$

$$K_{(\mathfrak{b}_2)k}(z,x) - K_{(\mathfrak{b}_1)k}(z,x) = (1 \otimes b_j)K_{(\mathfrak{b}_1)j(\mathfrak{b}_2)k}(z,x) - K_{(\mathfrak{b}_1)j(\mathfrak{b}_2)k}(z,x)(b_j \otimes 1), \tag{3.17c}$$

*where in each line there is an implicit summation over the repeated index $j$.*

*Proof.* By direct calculation, one can prove identity (3.17a):

$$
\begin{aligned}
K_{(\Delta\mathfrak{b}_{12})j(\mathfrak{b}_1)k}(z,x)(1 \otimes b_j) &= \sum_{j=1}^{h}\sum_{r,s\geq 0}\sum_{l=0}^{r}\omega_{(i_1\cdots i_l \shuffle p_1\cdots p_s)ji_{l+1}\cdots i_r k}(z,x)\, b_{i_1}\cdots b_{i_r} \otimes b_{p_1}\cdots b_{p_s}b_j \\
&= \sum_{r,s\geq 0}\sum_{l=0}^{r}\omega_{(i_1\cdots i_l \shuffle p_1\cdots p_s)p_{s+1}i_{l+1}\cdots i_r k}(z,x)\, b_{i_1}\cdots b_{i_r} \otimes b_{p_1}\cdots b_{p_s}b_{p_{s+1}} \\
&= \sum_{r,s\geq 0}\omega_{(i_1\cdots i_r \shuffle p_1\cdots p_s p_{s+1})k}(z,x)\, b_{i_1}\cdots b_{i_r} \otimes b_{p_1}\cdots b_{p_s}b_{p_{s+1}} \\
&= \sum_{\substack{r\geq 0 \\ s\geq 1}}\omega_{(i_1\cdots i_r \shuffle p_1\cdots p_s)k}(z,x)\, b_{i_1}\cdots b_{i_r} \otimes b_{p_1}\cdots \otimes b_{p_s} \\
&= K_{(\Delta\mathfrak{b}_{12})k}(z,x) - K_{(\mathfrak{b}_1)k}(z,x), 
\end{aligned}
\tag{3.18}
$$

where we have used the definition (3.7) together with eq. (3.15a) in the first line, relabeled $j \to p_{s+1}$ in the second line, got rid of the sum over $l$ by absorbing all indices in the shuffle-product in the third line and relabeled $s \to s+1$ in the fourth line. Finally, we used eq. (3.14) together with the definition (3.3) in the last line.

In a completely analogous way one can proof the contraction identities (3.17b) and (3.17c). $\qquad\square$

The identities (3.17) can be reformulated in terms of identities of Hopf algebra maps. While we refrain from showing them explicitly, since it is not very enlightening, this implies that the relations are merely consequences of combinatorics and not features of the component forms. In particular, these identities do not yield any interesting identities between the integration kernels. However, they will turn out to be useful tools for simplifying complicated expressions as for example eq. (6.11) when showing that eq. (4.11) reduces to the Fay identity at genus one.

For completing this technical section, let us note the echo of relations (2.7) for the integration kernels in terms of component forms. We find

$$K_{(\mathfrak{b})jk}(z,x) = K_{(\mathfrak{b})jk}(z,y), \quad \text{for } j \neq k, \tag{3.19a}$$

$$K_{(\mathfrak{b})jj'}(z,x)\big|_{j'=j} - K_{(\mathfrak{b})kk'}(z,x)\big|_{k'=k} = K_{(\mathfrak{b})jj'}(z,y)\big|_{j'=j} - K_{(\mathfrak{b})kk'}(z,y)\big|_{k'=k} \tag{3.19b}$$

for the component forms. Eqs. (3.19a),(3.19b) and (3.11) also imply the relation

$$K_{(\mathfrak{b})l}(y,z) - K_{(\mathfrak{b})l}(y,x) = \big(K_{(\mathfrak{b})ll'}(y,z) - K_{(\mathfrak{b})ll'}(y,x)\big)b_{l''}\Big|_{l''=l'=l}, \quad \text{for } l \in \{1,\ldots,h\}. \tag{3.20}$$

Tracing the above relations back to the corresponding kernel relations (2.7) follows immediately when performing an expansion order by order.

---

[9] Using the notation (3.16) the r.h.s. of eq. (3.17a) is abbreviated as

$$\mu_{12} \circ \mathcal{P}_{23}\big(K_{(\Delta\mathfrak{b}_{12})j(\mathfrak{b}_3)k}(z,x)\big) = \mu_{12}\big(K_{(\Delta\mathfrak{b}_{13})j(\mathfrak{b}_2)k}(z,x)\big) = K_{(\Delta\mathfrak{b}_{12})j(\mathfrak{b}_1)k}(z,x).$$

# 4 Identities for component forms

Employing the definitions and properties of the component forms introduced in § 3, we can now explore the space of possible identities. Linear 3-point identities have been shown to be trivially related to the kernel identities in eqs. (3.19) and (3.20). After stating a differential 2-point identity in § 4.1, we are going to consider quadratic 3-point identities for component forms in § 4.2. In § 5, we discuss the corresponding identities once expanded into integration kernels. The quadratic 3-point identities will be shown to qualify as valid generalizations of the genus-one Fay identities in § 6.

## 4.1 Differential 2-point identities

The first component form in Definition 3.3 takes values in the algebra $\mathfrak{b}$. As visible in eq. (3.3), the associated kernels are one-forms in the first variable and ordinary functions in the second variable. Consequently, one can apply the exterior derivative to the second variable of the first component form. Using the results from ref. [26, Lemma 8], one finds for the integration kernels

$$\mathrm{d}_x \omega_{i_1 \dots i_r jk}(z, x) = (-1)^r \mathrm{d}_z \omega_{i_r \dots i_1 jk}(x, z), \tag{4.1}$$

which has already been reported in [45, Corollary 9.5]. Written in terms of component forms

$$\mathrm{d}_x K_{(\mathfrak{b})jk}(z, x) = \mathrm{d}_z K_{(S\mathfrak{b})jk}(x, z) \tag{4.2}$$

it constitutes a neat example for the interplay between the two different notions of component forms. The above identity will be put to use in particular in § 7, where identities between higher-genus polylogarithms are considered.

## 4.2 Quadratic 3-point identities

In this subsection, we are going to identify quadratic combinations of component forms that depend on three points on the manifold, $\mathcal{Q}(z, y, x) \in \mathfrak{b} \otimes \mathfrak{b}$. These expressions will be one-forms in each of $z$ and $y$, and functions in $x$. Provided that $\mathcal{Q}$ vanishes for any combination of $z, y, x \in \Sigma$, and expanding $\mathcal{Q}$ in the generators of $\mathfrak{b} \otimes \mathfrak{b}$, one finds

$$0 \equiv \mathcal{Q}(z, y, x) = \sum_{r,s \geq 0} \kappa_{i_1 \cdots i_r, p_1 \cdots p_s}(z, y, x) \, b_{i_1} \cdots b_{i_r} \otimes b_{p_1} \cdots b_{p_s}, \tag{4.3a}$$

$$0 \equiv \kappa_{i_1 \cdots i_r, p_1 \cdots p_s}(z, y, x), \quad \forall i_1, \dots, i_r, p_1, \dots, p_s \in \{1, \dots, h\}, \tag{4.3b}$$

where the coefficients $\kappa$ are quadratic combinations of integration kernels from Enriquez' connection in eq. (2.3).

### 4.2.1 Index swapping identities

Starting from relations eq. (3.19) and eq. (3.20), one can derive so-called *index-swapping identities* relating first and second component forms. These identities will be used extensively throughout the later sections.

**Lemma 7.** *Let $k, l \in \{1, \ldots, h\}$ and consider the expression*

$$\mathcal{R}_{kl}(z, y, x) = K_{(\mathfrak{b}_1)j}(z, x)\Big(K_{(\mathfrak{b}_2)jk}(y, z) - K_{(\mathfrak{b}_2)jk}(y, x) - \delta_{jk}K_{(\mathfrak{b}_2)lm}(y, z) + \delta_{jk}K_{(\mathfrak{b}_2)lm}(y, x)\Big)\Big|_{m=l}. \tag{4.4}$$

*Then $\mathcal{R}_{kl}(z, y, x) = 0$ for all pairwise distinct points $z, y, x \in \Sigma$.*

*Proof.* We start with the expression $K_{(\mathfrak{b}_1)j}(z, x)\left[K_{(\mathfrak{b}_2)jk}(y, z) - K_{(\mathfrak{b}_2)jk}(y, x)\right]$ and use eq. (3.19a) to conclude that

$$K_{(\mathfrak{b}_1)j}(z, x)\left[K_{(\mathfrak{b}_2)jk}(y, z) - K_{(\mathfrak{b}_2)jk}(y, x)\right] = K_{(\mathfrak{b}_1)j}(z, x)\delta_{jk}\left[K_{(\mathfrak{b}_2)km}(y, z) - K_{(\mathfrak{b}_2)km}(y, x)\right]\Big|_{m=k}. \tag{4.5}$$

As a next step, we use eq. (3.19b) to arrive at

$$K_{(\mathfrak{b}_1)j}(z, x)\left[K_{(\mathfrak{b}_2)jk}(y, z) - K_{(\mathfrak{b}_2)jk}(y, x)\right] = K_{(\mathfrak{b}_1)j}(z, x)\delta_{jk}\left[K_{(\mathfrak{b}_2)lm}(y, z) - K_{(\mathfrak{b}_2)lm}(y, x)\right]\Big|_{m=l}, \tag{4.6}$$

which immediately yields $\mathcal{R}_{kl}(z, y, x) = 0$. $\qquad\square$

We will also need a similar index swapping identity for component forms involving the coproduct.

**Lemma 8.** *Let $k \in \{1, \ldots, h\}$ and define*

$$\begin{aligned}
\mathcal{R}_k(z, y, x) =&\left(K_{(\mathfrak{b}_2)j}(y, z) - K_{(\mathfrak{b}_2)j}(y, x)\right)K_{(\mathfrak{b}_1)j(\Delta\mathfrak{b}_{12})k}(z, x) \\
&- \left(K_{(\mathfrak{b}_2)jk}(y, z) - K_{(\mathfrak{b}_2)jk}(y, x)\right)\left(K_{(\Delta\mathfrak{b}_{12})j}(z, x) - K_{(\mathfrak{b}_1)j}(z, x)\right).
\end{aligned} \tag{4.7}$$

*Then $\mathcal{R}_k(z, y, x) = 0$ for all pairwise distinct $z, y, x \in \Sigma$.*

*Proof.* We can apply eqs. (3.19b) and (3.20) to obtain

$$\begin{aligned}
&\left[K_{(\mathfrak{b}_2)j}(y, z) - K_{(\mathfrak{b}_2)j}(y, x)\right]K_{(\mathfrak{b}_1)j(\Delta\mathfrak{b}_{12})k}(z, x) \\
&\qquad\qquad = \left[K_{(\mathfrak{b}_2)ll'}(y, z) - K_{(\mathfrak{b}_2)ll'}(y, x)\right]\Big|_{l'=l}(1 \otimes b_j)K_{(\mathfrak{b}_1)j(\Delta\mathfrak{b}_{12})k}(z, x)
\end{aligned} \tag{4.8}$$

for any $l \in \{1, \ldots, h\}$. On the other hand, we can use eq. (3.19a) and eq. (3.19b) to conclude

$$\begin{aligned}
&\left[K_{(\mathfrak{b}_2)jk}(y, z) - K_{(\mathfrak{b}_2)jk}(y, x)\right]\left[K_{(\Delta\mathfrak{b}_{12})j}(z, x) - K_{(\mathfrak{b}_1)j}(z, x)\right] \\
&\qquad\qquad = \left[K_{(\mathfrak{b}_2)ll'}(y, z) - K_{(\mathfrak{b}_2)ll'}(y, x)\right]\Big|_{l'=l}\left[K_{(\Delta\mathfrak{b}_{12})k}(z, x) - K_{(\mathfrak{b}_1)k}(z, x)\right]. \tag{4.9}
\end{aligned}$$

Subtracting the above two equations from each other yields

$$\begin{aligned}
&\mathcal{R}_k(z, y, x) \\
&= \left[K_{(\mathfrak{b}_2)ll'}(y, z) - K_{(\mathfrak{b}_2)ll'}(y, x)\right]\Big|_{l'=l}\left[(1 \otimes b_j)K_{(\mathfrak{b}_1)j(\Delta\mathfrak{b}_{12})k}(z, x) - \left(K_{(\Delta\mathfrak{b}_{12})k}(z, x) - K_{(\mathfrak{b}_1)k}(z, x)\right)\right],
\end{aligned} \tag{4.10}$$

where we conclude that the r.h.s. does indeed vanish by means of the contraction identity (3.17b), which completes the proof. $\qquad\square$

Notice that these identities are trivial in the sense that they follow directly from the properties (3.19) and (3.20). However, the form derived here will prove very useful for transforming identities in later sections.

### 4.2.2 The quadratic Fay-like identity

In this section, we state and prove a quadratic 3-point identity among the component forms. We call these identities *Fay-like*, which is supported by the fact that we show in §6 that this identity reduces to the well-known Fay identity for the elliptic Kronecker function upon restriction to genus one. However, in contrast to the elliptic case, it is currently not clear if and how this generalized Fay-like identity may connect to the Fay trisecant equation for higher-genus theta functions [10, 11].

**Theorem 9** (Fay-like identity). *Let $k, l, m \in \{1, \ldots, h\}$ and $K_{(\mathfrak{b})j}(z, x)$, $K_{(\mathfrak{b}_1)j(\mathfrak{b}_2)k}(z, x)$ be the first and second component form as in Definitions 4 and 5, respectively. Define*

$$\mathcal{Q}_{klm}(z, y, x) = q_{klm}(z, y, x) + \mathcal{P}_{12}(q_{kml}(y, z, x)). \tag{4.11}$$

*where*

$$
\begin{aligned}
q_{klm}(z, y, x) = & \left[ K_{(\mathfrak{b}_1)l}(z, x) - K_{(\mathfrak{b}_1)l}(z, y) \right] K_{(\mathfrak{b}_2)k}(y, x)(1 \otimes b_m) \\
& - K_{(\mathfrak{b}_1)j}(z, y) K_{(\mathfrak{b}_2)j(\Delta \mathfrak{b}_{12})k}(y, x)(b_l \otimes b_m) .
\end{aligned}
\tag{4.12}
$$

*Then $\mathcal{Q}_{klm}(z, y, x) = 0$ for all pairwise distinct points $z, y, x \in \Sigma$.*

*Proof.* The first step of the proof is to study the analytical behavior of $\mathcal{Q}_{klm}$. In Appendix A.2, we show explicitly that we have

$$\mathcal{Q}_{klm}(\gamma_i z, y, x) = (e^{b_i} \otimes 1)\mathcal{Q}_{klm}(z, y, x), \quad \mathcal{Q}_{klm}(z, \gamma_i y, x) = (1 \otimes e^{b_i})\mathcal{Q}_{klm}(z, y, x), \tag{4.13}$$

and that the combination of terms is holomorphic in the variables $z$, $y$ and $x$.

Suppose we integrate with respect to the variable $z$, defining a one-form $\Omega_{klm}(z, y, x)$,

$$\Omega_{klm}(z, y, x) = \int_{z'=z_0}^{z} \mathcal{Q}_{klm}(z', y, x), \tag{4.14}$$

where $z_0$ is an arbitrary fixed point on the surface, and the integral is homotopy-invariant since $\mathcal{Q}_{klm}$ is holomorphic in $z$. Fixing arbitrary $z$ and $x$, we can interpret $\Omega_{klm}(z, y, x)$ as a holomorphic one-form satisfying

$$\Omega_{klm}(z, \gamma_i y, x) = (1 \otimes e^{b_i})\Omega_{klm}(z, y, x). \tag{4.15}$$

The quasi-periodicity and (vanishing) residue of $\Omega_{klm}$ are similar to the conditions that had uniquely determined Enriquez' connection in eq. (2.2). Indeed, as detailed in Appendix A.3, the strategy used to uniquely identifiy Enriquez' connection also allows us allow us to uniquely identify $\Omega_{klm}(z, y, x) \equiv 0$. We can reverse the integration in eq. (4.14) by taking an exterior derivative, finding

$$\mathcal{Q}_{klm}(z, y, x) = \mathrm{d}_z(\Omega_{klm}(z, y, x)) \equiv 0. \tag{4.16}$$

$\square$

### 4.2.3 Identities for $z$-reduction

In the process of rewriting scattering amplitudes and other observables in the language of polylogarithms formally defined in eq. (2.6), one frequently faces the situation that an integration variable will occur multiple times as argument of two integration kernels, for example:

$$\omega_{i_1 \cdots i_r}(z, x)\omega_{p_1 \cdots p_s}(y, z) \tag{4.17}$$

In order to (at least formally) perform the next integration, one will have to rewrite the product of the two kernels as

$$\omega_{i_1\cdots i_r}(z,x)\,\omega_{p_1\cdots p_s}(y,z) = \sum_{r',s'} \lambda_{i'_1\cdots i'_{r'},p'_1\cdots p'_{s'}}\omega_{i'_1\cdots i'_{r'}}(z,x)\,\omega_{p'_1\cdots p'_{s'}}(y,x)$$
$$+\eta_{i'_1\cdots i'_{r'},p'_1\cdots p'_{s'}}\omega_{i'_1\cdots i'_{r'}}(z,x)\,\omega_{p'_1\cdots p'_{s'}}(y,z) \tag{4.18}$$

with some constant coefficients $\lambda_{\cdots}$ and $\eta_{\cdots}$ contracted with the terms on the r.h.s., with an implicit dependence on the indices $i_1\cdots i_r, p_1\cdots p_s$. This process has been referred to as "removing repeated occurrences of $z$" [4] and – more recently – "$z$-reduction" in ref. [45]. In a genus-one context the Fay identities were sufficient to $z$-reduce all occurring expressions. Similarly, we will employ the Fay-like identities from Theorem 9 to perform this task for the kernels derived from Enriquez connection at higher genera.

We are going to rewrite identity eq. (4.11) from Theorem 9 in order to isolate a single term with repeated dependence on the argument $z$. In particular, this means that we want to collapse the last line of eq. (4.11) into a single term. To achieve this, we apply the index swapping identities from § 4.2.1.

Once the term with repeated $z$-dependence has been isolated, we can perform a relabeling of the $b$-alphabet in order to arrive at an equivalent identity, which exactly matches the identity stated in [45, Conjecture 9.7] upon expansion into kernels. The result in this section therefore proves this conjecture and emphasizes the benefits of the formalism developed in this article.

**Isolating repeated $z$-dependence.**  Notice, that the Fay-like identity (4.11) can be factorized by means of eq. (3.19b) and eq. (3.20) to yield an identity with a factor $b_l \otimes b_m$

$$\mathcal{Q}'_{klm}(z,y,x) = \Big( \big[ (K_{(\mathfrak{b}_1)pp'}(z,x) - K_{(\mathfrak{b}_1)pp'}(z,y)) \big] \Big|_{p'=p} K_{(\mathfrak{b}_2)k}(y,x)$$
$$+ \big[ (K_{(\mathfrak{b}_2)qq'}(y,x) - K_{(\mathfrak{b}_2)qq'}(y,z)) \big] \Big|_{q'=q} K_{(\mathfrak{b}_1)k}(z,x) \tag{4.19}$$
$$- K_{(\mathfrak{b}_1)j}(z,y)K_{(\mathfrak{b}_2)j(\Delta\mathfrak{b}_{12})k}(y,x) - K_{(\mathfrak{b}_2)j}(y,z)K_{(\mathfrak{b}_1)j(\Delta\mathfrak{b}_{12})k}(z,x) \Big) (b_l \otimes b_m),$$

for arbitrary $p,q \in \{1,\ldots,h\}$. Applying now the index swapping identities (4.4) and (4.7) leads to an expression with a single isolated term with repeated $z$-dependence (see Appendix A.4 for more details of the derivation)

$$\mathcal{Q}''_{klm}(z,y,x) = \Big( \big[ K_{(\mathfrak{b}_2)jk}(y,x) - K_{(\mathfrak{b}_2)jk}(y,z) \big] K_{(\Delta\mathfrak{b}_{12})j}(z,x)$$
$$+ \big[ K_{(\mathfrak{b}_1)jk}(z,x) - K_{(\mathfrak{b}_1)jk}(z,y) \big] K_{(\mathfrak{b}_2)j}(y,x)$$
$$- K_{(\mathfrak{b}_2)j}(y,x)K_{(\mathfrak{b}_1)j(\Delta\mathfrak{b}_{12})k}(z,x)) - K_{(\mathfrak{b}_1)j}(z,y)(K_{(\mathfrak{b}_2)j(\Delta\mathfrak{b}_{12})k}(y,x) \Big) (b_l \otimes b_m)$$
$$=: \mathcal{I}_k(z,y,x)(b_l \otimes b_m) . \tag{4.20}$$

**An equivalent identity.**  In the following, we would like to rewrite identity (4.20) to allow us to directly match – upon expansion – against the identity in ref. [45, Conjecture 9.7]. The rewriting shall be implemented by defining a map $\Xi$ to be applied to eq. (4.20). In order to construct the mapping, it should be noted that transformations of algebra generators do not affect which arguments a generating function uses. Given that both identities have precisely one term with repeated $z$-dependence, it should be possible to construct the map from demanding they are mapped onto each other.

Accordingly, we want to transform the algebra generators such that the mapping

$$K_{(\mathfrak{b}_2)jk}(y,z)K_{(\Delta\mathfrak{b}_{12})j}(z,x) \mapsto K_{(\mathfrak{b}_1)j}(z,x)K_{(S\mathfrak{b}_2)jk}(y,z) \tag{4.21}$$

is realized for the term with repeated $z$-dependence, which can be achieved employing the Hopf algebra structure.

The notation in eq. (4.21) already suggests that the transformation should somehow correspond to first applying the antipode to the second tensor site and applying the coproduct to the first site in a second step. The application of the antipode is motivated by the fact that the first and second component forms should swap upon application of the transformation (4.21) and also by the appearance of the antipode in the final expression. The application of the coproduct is suggested by the requirement to remove the coproduct from the final expression. We now want to formally define a map that realizes the transformation (4.21).

**Definition 10.** *We define the map* $\Xi : \mathfrak{b}^{\otimes 2} \to \mathfrak{b}^{\otimes 2}$ *as the combination*

$$\Xi = \mu_{23} \circ \Delta_1 \circ S_2 \,, \tag{4.22}$$

*where the subscripts as usual indicate the tensor sites on which the maps are applied.*

As shown in Appendix A.4, the map $\Xi$ exactly realizes the desired transformation (4.21). Applying the map $\Xi$ to the quantity $\mathcal{I}_k$ defined in eq. (4.20), we obtain

$$
\begin{aligned}
\tilde{\mathcal{I}}_k(z,y,x) &= \Xi \circ \mathcal{I}_k(z,y,x), \\
&= \left[ K_{(S\mathfrak{b}_2)jk}(y,x) - K_{(S\mathfrak{b}_2)jk}(y,z) \right] K_{(\mathfrak{b}_1)j}(z,x) \\
&\quad + \left[ K_{(\Delta\mathfrak{b}_{12})jk}(z,x) - K_{(\Delta\mathfrak{b}_{12})jk}(z,y) \right] K_{(S\mathfrak{b}_2)j}(y,x) \\
&\quad - K_{(\Delta\mathfrak{b}_{12})j(\mathfrak{b}_1)k}(z,x) K_{(S\mathfrak{b}_2)j}(y,x) - K_{(\Delta\mathfrak{b}_{12})j}(z,y) K_{(S\mathfrak{b}_2)j(\mathfrak{b}_1)k}(y,x) \,.
\end{aligned}
\tag{4.23}
$$

While eq. (4.23) is already in beautiful shape for removing double occurrences of $z$, let us rewrite the identity one last time by removing contractions of indices in the first line by using eqs. (3.19a) and (3.19b):

$$
\begin{aligned}
\tilde{\mathcal{I}}'_k(z,y,x) &= \left[ K_{(S\mathfrak{b}_2)pp'}(y,x) - K_{(S\mathfrak{b}_2)pp'}(y,z) \right] \Big|_{p'=p} K_{(\mathfrak{b}_1)k}(z,x) \\
&\quad + \left[ K_{(\Delta\mathfrak{b}_{12})jk}(z,x) - K_{(\Delta\mathfrak{b}_{12})jk}(z,y) \right] K_{(S\mathfrak{b}_2)j}(y,x) \\
&\quad - K_{(\Delta\mathfrak{b}_{12})j(\mathfrak{b}_1)k}(z,x) K_{(S\mathfrak{b}_2)j}(y,x) - K_{(\Delta\mathfrak{b}_{12})j}(z,y) K_{(S\mathfrak{b}_2)j(\mathfrak{b}_1)k}(y,x) \,.
\end{aligned}
\tag{4.24}
$$

where the above identity is valid for any values of $p = 1, \ldots, h$. Using the eq. (4.24), we will straightforwardly be able to extract kernel identities in the next section.

## 5  Identities for integration kernels

The identities formulated in the previous subsection in terms of component forms provide a convenient way of obtaining relations between the integration kernels for higher-genus polylogarithms: they just need to be expanded in algebra generators. In this section, we aim to make this explicit and thereby state a collection of useful identities between integration kernels, which can then be used to derive functional relations between higher-genus polylogarithms in § 7.

Let us expand eq. (4.24) into component forms. While the explicit calculation is performed in Appendix A.4, the resulting identity arising as coefficient corresponding to a word

$(-1)^s b_{i_1} \ldots b_{i_r} \otimes b_{p_s} \ldots b_{p_1}$ reads

$$\omega_{i_1 \ldots i_r k}(z,x)\, \omega_{p_1 \ldots p_s p p'}(y,z)\Big|_{p'=p}$$

$$= \omega_{i_1 \ldots i_r k}(z,x)\, \omega_{p_1 \ldots p_s p p'}(y,x)\Big|_{p'=p}$$

$$- \sum_{l=0}^{r-1} \sum_{m=0}^{s} (-1)^{m-s} \omega_{(i_1 \ldots i_l ⧢ p_s \ldots p_{m+1}) j i_{l+1} \ldots i_r k}(z,x)\, \omega_{p_1 \ldots p_m j}(y,x)$$

$$- \sum_{m=0}^{s} (-1)^{m-s} \omega_{(i_1 \ldots i_r ⧢ p_s \ldots p_{m+1}) jk}(z,y)\, \omega_{p_1 \ldots p_m j}(y,x)$$

$$- \sum_{l=0}^{r} \sum_{m=0}^{s} (-1)^{m-s} \omega_{(i_1 \ldots i_l ⧢ p_s \ldots p_{m+1}) j}(z,y)\, \omega_{p_1 \ldots p_m j i_{l+1} \ldots i_r k}(y,x)$$

$$=: \mathcal{U}_{i_1 \ldots i_r k, p_1 \ldots p_s p}(z,y,x)$$

$$(5.1)$$

and is valid for each set of indices $(i_1, \ldots, i_r, k, p_1, \ldots, p_s, p) \in \{1, \ldots, h\}^{r+s+2}$. In the above equation, $\mathcal{U}_{i_1 \ldots i_r k, p_1 \ldots p_s p}(z,y,x)$ denotes the collection of all $z$-reduced terms.

Notice that eq. (5.1) matches the identity stated in ref. [45, Conjecture 9.7] upon application of the expanded index swapping identity from Lemma 7 and absorption of the index $j$ into the shuffle product in the second line. Furthermore, the interchange lemmas from ref. [45, Theorem 9.2, Corollary 9.3] can be shown to be equivalent to eq. (5.1) in the special case of $r = 0$ by virtue of Lemma 7 and some combinatorics.

## 5.1 Uniqueness of identities for kernels

In this subsection we prove that the Fay-like identity (4.11) is the unique quadratic identity up to linear combinations of the trivial linear identities (3.19a) and (3.19b). Following the strategy from § 2.1, we prove this statement using expansions of the component forms in eq. (4.11) into integration kernels. It is currently unsettled, whether the uniqueness statement can be lifted to a statement on identities between generating functions consistently. The uniqueness statement implies that the collection of conjectures in [45] is complete.

Proving uniqueness proceeds in two steps: we first notice that relation eq. (5.1) together with the linear identity (2.7a) allows to find a $z$-reduced expression for any quadratic combination of the integration kernels with repeated $z$-dependence. Afterwards, this fact can be combined with the following lemma to deduce the uniqueness in Theorem 12 below.

**Lemma 11** (Uniqueness of $z$-reduced quadratic identities). *Any $z$-reduced 3-point quadratic identity holds by means of the trivial linear equations (2.7a) and (2.7b). Moreover, it does so separately for each weight implying a weight-grading of $z$-reduced quadratic identities.*

*Proof.* Consider an arbitrary $z$-reduced quadratic identity of highest weight $m$,

$$\mathcal{I}^{(m)}(z,y,x) = \sum_{r=0}^{m} \sum_{s=0}^{m-r} [\lambda_{i_1 \ldots i_r i, p_1 \ldots p_s p}\, \omega_{i_1 \ldots i_r i}(z,x) + \eta_{i_1 \ldots i_r i, p_1 \ldots p_s p}\, \omega_{i_1 \ldots i_r i}(z,y)]\, \omega_{p_1 \ldots p_s p}(y,x) = 0$$

$$(5.2)$$

whose vanishing is achieved by a particular fixed choice of coefficients $\lambda_{i_1 \ldots i_r i, p_1 \ldots p_s p}, \eta_{i_1 \ldots i_r i, p_1 \ldots p_s p} \in \mathbb{C}$. We will show that each term in the sum of eq. (5.2) vanishes due to the trivial linear identities (2.7) by induction.

**Induction start:** we will show that the terms with $r = m$ in eq. (5.2) vanish trivially. Similar to the linear case (cf. eq. (2.9)), we define a family of lower weight identities by iteratively applying the

quasi-periodicity rule first in $z$ and then in $y$.

$$\mathcal{I}_{i_1\cdots i_k}^{(m-k,0)}(z,y,x) := \mathcal{I}_{i_1\cdots i_{k-1}}^{(m-k+1,0)}(\gamma_{i_k}z,y,x) - \mathcal{I}_{i_1\cdots i_{k-1}}^{(m-k+1,0)}(z,y,x) = 0, \tag{5.3a}$$

$$\mathcal{I}_{i_1\cdots i_{k+l}}^{(m-k-l,l)}(z,y,x) := \mathcal{I}_{i_1\cdots i_{k+l-1}}^{(m-k-l+1,l-1)}(z,\gamma_{i_{k+l}}y,x) - \mathcal{I}_{i_1\cdots i_{k+l-1}}^{(m-k-l+1,l-1)}(z,y,x) = 0, \tag{5.3b}$$

for arbitrary choices of $k, l \geq 0$ subject to $1 \leq k + l \leq m$ and $i_1, \ldots, i_{k+l} \in \{1, \ldots, h\}$ and with the identification $\mathcal{I}^{(m,0)} := \mathcal{I}^{(m)}$. Notice that $\mathcal{I}_{i_1\cdots i_{k+l}}^{(m-k-l,l)}(z,y,x)$ indeed has lower weight as the highest weight terms are canceled out.

At first, let us consider $\mathcal{I}_{i_1\cdots i_k}^{(0,0)}(z,y,x)$, where only the quasi-periodicity in $z$ is used $m$ times. Since the quasi-periodicity in $z$ only affects the terms in square brackets in eq. (5.2), when we apply the quasi-periodicity $m$ times, the non-zero contribution will only come from the term with $r = m$ and $s = 0$ in eq. (5.2), because application of quasi-periodicity in $z$ $m$ times will kill $m$ first indices in $\omega_{i_1\cdots i_r i}(z, \cdot)$. Therefore, it is easy to calculate

$$\mathcal{I}_{i_1\cdots i_m}^{(0,0)}(z,y,x) = [\lambda_{i_1\cdots i_m i,p}\,\omega_i(z) + \eta_{i_1\cdots i_m i,p}\,\omega_i(z)]\,\omega_p(y) = 0. \tag{5.4}$$

which directly implies

$$\lambda_{i_1\cdots i_m i,p} + \eta_{i_1\cdots i_m i,p} = 0. \tag{5.5}$$

Now, we consider the identity $\mathcal{I}_{i_1\cdots i_{m-1}}^{(1,0)}(z,y,x)$, where $z$ quasi-periodicity is applied $(m-1)$ times. The non-trivial contribution to $\mathcal{I}_{i_1\cdots i_{m-1}}^{(1,0)}(z,y,x)$ will come from the terms $r = m, m-1$ and $s = 0, \ldots, m-r$ in eq. (5.2). Evidently, the contribution from $r = m-1$ terms will be holomorphic, because application of quasi-periodicity in $z$ $(m-1)$ times will kill all but one index in $\omega_{i_1\cdots i_r i}(z, \cdot)$. Therefore, it is easy to verify that

$$\mathcal{I}_{i_1\cdots i_{m-1}}^{(1,0)}(z,y,x) = [\lambda_{i_1\cdots i_m i,p}\,\omega_{i_m i}(z,x) + \eta_{i_1\cdots i_m i,p}\,\omega_{i_m i}(z,y)]\,\omega_p(y) + [\text{regular in } z] = 0. \tag{5.6}$$

Taking residues at $z = x$ or $z = y$ allows us to isolate terms with $\lambda_{\ldots}$ and $\eta_{\ldots}$, resulting in the relations

$$\operatorname*{Res}_{z=x}\mathcal{I}_{i_1\cdots i_{m-1}}^{(1,0)}(z,y,x) = 0 \implies \lambda_{i_1\cdots i_{m-1} ii,p} = 0, \tag{5.7a}$$

$$\operatorname*{Res}_{z=y}\mathcal{I}_{i_1\cdots i_{m-1}}^{(1,0)}(z,y,x) = 0 \implies \eta_{i_1\cdots i_{m-1} ii,p} = 0. \tag{5.7b}$$

Equations (5.5) and (5.7) together with the trivial linear identities (2.7) now yield

$$\begin{aligned}
[\lambda_{i_1\cdots i_m i,p}\,\omega_{i_1\cdots i_m i}(z,x) &+ \eta_{i_1\cdots i_m i,p}\,\omega_{i_1\cdots i_m i}(z,y)]\,\omega_p(y) \\
&= \lambda_{i_1\cdots i_m i,p}[\omega_{i_1\cdots i_m i}(z,x) - \omega_{i_1\cdots i_m i}(z,y)]\,\omega_p(y) \\
&= \lambda_{i_1\cdots ii,p}[\omega_{i_1\cdots kk'}(z,x) - \omega_{i_1\cdots kk'}(z,y)]\Big|_{k'=k}\,\omega_p(y) = 0,
\end{aligned} \tag{5.8}$$

for some fixed $1 \leq k \leq h$. We have therefore shown that the $r = m$ term in eq. (5.2) indeed trivially vanishes, thus concluding the induction start.

**Induction step:** assume that all terms with $r > m - k$ in eq. (5.2) trivially vanish, so that it can be written as

$$\mathcal{I}^{(m)}(z,y,x) = \sum_{r=0}^{m-k}\sum_{s=0}^{m-r}[\lambda_{i_1\cdots i_r i, p_1\cdots p_s p}\,\omega_{i_1\cdots i_r i}(z,x) + \eta_{i_1\cdots i_r i, p_1\cdots p_s p}\,\omega_{i_1\cdots i_r i}(z,y)]\,\omega_{p_1\cdots p_s p}(y,x) = 0 \tag{5.9}$$

Now, we would like to show that the terms with $r = m - k$ vanish trivially too.

To begin with, we consider the identity $\mathcal{I}_{i_1\cdots i_{m-k}}^{(k,0)}(z,y,x)$. Similarly to the argumentation at the induction start, the non-trivial contribution after applying $z$ quasi-periodicity $(m-k)$ times comes

from the terms with $r = m - k$, leading to

$$\mathcal{I}^{(k,0)}_{i_1\cdots i_{m-k}}(z,y,x) = \sum_{s=0}^{k} \left[\lambda_{i_1\cdots i_m i, p_1\cdots p_s p}\,\omega_i(z) + \eta_{i_1\cdots i_m i, p_1\cdots p_s p}\,\omega_i(z)\right]\omega_{p_1\cdots p_s p}(y,x) = 0. \tag{5.10}$$

Now we can apply the $y$ quasi-periodicity $k$ times, which eliminates all terms in eq. (5.10) with $s < k$ leading to

$$\mathcal{I}^{(0,k)}_{i_1\cdots i_{m-k}p_1\cdots p_k}(z,y,x) = \left[\lambda_{i_1\cdots i_m i, p_1\cdots p_k p}\,\omega_i(z) + \eta_{i_1\cdots i_m i, p_1\cdots p_k p}\,\omega_i(z)\right]\omega_p(y)$$

$$\implies \lambda_{i_1\cdots i_{m-k} i, p_1\cdots p_k p} + \eta_{i_1\cdots i_{m-k} i, p_1\cdots p_k p} = 0. \tag{5.11}$$

Inserting this relation back into eq. (5.10) eliminates the terms with $s = k$ and we can now take the $y$ quasi-periodicity $(k-1)$ times and repeat the argument. Therefore, we conclude that

$$\lambda_{i_1\cdots i_{m-k} i, p_1\cdots p_s p} + \eta_{i_1\cdots i_{m-k} i, p_1\cdots p_s p} = 0 \tag{5.12}$$

for all $s = 1, \ldots, k$.

As at the induction start, let us now consider the application of $z$ quasi-periodicity $(m-k-1)$ times. The non-trivial non-holomorphic contribution will come from the terms with $r = m - k$ in eq. (5.9), thus allowing us to calculate

$$\mathcal{I}^{(k+1,0)}_{i_1\cdots i_{m-k-1}}(z,y,x) = \sum_{s=0}^{k} \left[\lambda_{i_1\cdots i_{m-k} i, p_1\cdots p_s p}\,\omega_{i_{m-k} i}(z,x) + \eta_{i_1\cdots i_{m-k} i, p_1\cdots p_s p}\,\omega_{i_{m-k} i}(z,y)\right]\omega_{p_1\cdots p_s p}(y,x)$$

$$+ \left[\text{regular in } z\right] = 0. \tag{5.13}$$

Now we can take residue of eq. (5.13) at $z = y$ or $z = x$, that gives us

$$\operatorname*{Res}_{z=x}\mathcal{I}^{(k+1,0)}_{i_1\cdots i_{m-k-1}}(z,y,x) = \sum_{s=0}^{k} \lambda_{i_1\cdots i_{m-k-1} i i, p_1\cdots p_s p}\omega_{p_1\cdots p_s p}(y,x) = 0, \tag{5.14a}$$

$$\operatorname*{Res}_{z=y}\mathcal{I}^{(k+1,0)}_{i_1\cdots i_{m-k-1}}(z,y,x) = \sum_{s=0}^{k} \eta_{i_1\cdots i_{m-k-1} i i, p_1\cdots p_s p}\omega_{p_1\cdots p_s p}(y,x) = 0. \tag{5.14b}$$

The r.h.s. of eq. (5.14) is a one-form in $y$ and we can take quasi-periodicity $k$ times in order to eliminate all terms in the sums of eq. (5.14) but those with $s = k$. This gives us that $\lambda_{i_1\cdots i_{m-k-1} i i, p_1\cdots p_k p} = \eta_{i_1\cdots i_{m-k-1} i i, p_1\cdots p_k p} = 0$ and we can insert this result back to eq. (5.14). Repeating this argument $k$ times allows us to conclude that

$$\lambda_{i_1\cdots i_{m-k-1} i i, p_1\cdots p_s p} = \eta_{i_1\cdots i_{m-k-1} i i, p_1\cdots p_s p} = 0 \tag{5.15}$$

for all $s = 1, \ldots, k$.

Using the results (5.11) and (5.15) allows us to show that for $r = m - k$ and each $s = 0, \ldots, k$ we have

$$\left[\lambda_{i_1\cdots i_{m-k} i, p_1\cdots p_s p}\,\omega_{i_1\cdots i_{m-k} i}(z,x) + \eta_{i_1\cdots i_{m-k} i, p_1\cdots p_s p}\,\omega_{i_1\cdots i_{m-k} i}(z,y)\right]\omega_{p_1\cdots p_s p}(y,x)$$

$$= \lambda_{i_1\cdots i_{m-k} i, p_1\cdots p_s p}\left[\omega_{i_1\cdots i_{m-k} i}(z,x) - \omega_{i_1\cdots i_{m-k} i}(z,y)\right]\omega_{p_1\cdots p_s p}(y,x)$$

$$= \lambda_{i_1\cdots i_{m-k-1} i i, p}\left[\omega_{i_1\cdots i_{m-k-1} k k'}(z,x) - \omega_{i_1\cdots i_{m-k-1} k k'}(z,y)\right]\Big|_{k'=k}\omega_{p_1\cdots p_s p}(y,x)$$

$$= 0, \tag{5.16}$$

by means of the trivial identities (2.7). This concludes the proof. $\qquad\square$

The above lemma proves that no non-trivial identities among $z$-reduced terms exist (of which weight grading is an immediate consequence). This allows us to conclude that the Fay-like

identity (4.11) expanded into kernels is the unique quadratic 3-point identity:

**Theorem 12** (Uniqueness of the Fay-like identity). *Any quadratic 3-point identity among the integration kernels is equivalent to a linear combination of the Fay-like identity (5.1) and the trivial linear identities (2.7a) and (2.7b), meaning that the Fay-like identity (4.11) is unique modulo the trivial linear identities.*

*Proof.* Let $\mathcal{I}(z, y, x) = 0$ be an arbitrary 3-point identity among integration kernels such that $\mathcal{I}(z, y, x)$ is a 1-form in $z, y$ and a function in $x$. We begin by grouping the terms in $\mathcal{I}(z, y, x) = 0$ such that

$$\mathcal{I}(z, y, x) = 0 \Leftrightarrow \sum_{r=0}^{m} \sum_{s=0}^{m-r} \nu_{i_1 \cdots i_r i, p_1 \cdots p_s p}\, \omega_{i_1 \cdots i_r i}(z, x)\, \omega_{p_1 \cdots p_s p}(y, z) = \mathcal{Z}(z, y, x)\,, \tag{5.17}$$

where $\mathcal{Z}(z, y, x)$ is a placeholder function that collects all $z$-reduced terms, $m$ is the weight of the highest-weight term, and $\nu_{i_1 \cdots i_r i, p_1 \cdots p_s p} \in \mathbb{C}$ are arbitrary coefficients defined by the original identity. Using eq. (2.7a) for $p_s \neq p$ and eq. (5.1) for $p_s = p$ to simplify the r.h.s., one finds

$$\sum_{r=0}^{m} \sum_{s=0}^{m-r} \nu_{i_1 \cdots i_r i, p_1 \cdots p_s p}\, \omega_{i_1 \cdots i_r i}(z, x)\, \omega_{p_1 \cdots p_s p}(y, z)$$

$$= \sum_{r=0}^{m} \sum_{s=0}^{m-r} \left[\nu_{i_1 \cdots i_r i, p_1 \cdots p_s p}\, \omega_{i_1 \cdots i_r i}(z, x)\, \omega_{p_1 \cdots p_s p}(y, x)\right]\Big|_{p \neq p_s} \tag{5.18}$$

$$+ \left[\nu_{i_1 \cdots i_r i, p_1 \cdots p_s p}\, \mathcal{U}_{i_1 \cdots i_r i, p_1 \cdots p_{s-1} p}(z, y, x)\right]\Big|_{p = p_s}$$

$$=: \mathcal{Z}_{\mathrm{F}}(z, y, x)$$

where $\mathcal{Z}_{\mathrm{F}}(z, y, x)$ is a placeholder function that collects the $z$-reduced terms coming from the trivial and Fay-like identity. Then, the difference $\mathcal{Z}(z, y, x) - \mathcal{Z}_{\mathrm{F}}(z, y, x) = 0$ is a $z$-reduced quadratic identity, which holds by means of the trivial linear equations (2.7a) and (2.7b) as a result of Lemma 11. Thus, we recognize that the identity $\mathcal{I}(z, y, x) = 0$ is equivalent to a linear combination of eqs. (2.7a), (2.7b), and (5.1) as desired. □

As a final remark, the identity given in ref. [45, Conjecture 9.6], which superficially depends on arbitrary many points on the Riemann surface, can be reduced to the proven identity eq. (5.1). One has to notice that the dependence on the points $a_l$ and $b_l$ can be eliminated by means of the linear relations eqs. (2.7). Then the identity only depends on three points and Theorem 12 can be applied.

# 6 Comparison to generating functions at genus one

Enriquez' generating function eq. (2.3) is the unique solution to two conditions: while the pole structure (with corresponding residues) is fixed by the (analytic) condition (2.2b), its periodicities are constrained algebraically through (2.2a). When constructing Fay-like identities as above, residues at the poles have to add to zero and periodicities have to match.

The full range of analytic and non-abelian algebraic structures can only be implemented for genus two and higher. In this section we are going to explore what remains from the analytic and algebraic structures when constructing Fay-like identities at genera zero and one.

## 6.1 Echo of the formalism at genus zero

At genus zero, there are no non-trivial cycles on the Riemann sphere. As all generators of the algebra correspond to "moving around a cycle of the Riemann surface", there are no generators

and the algebra is trivial. Accordingly, there are no algebraic (quasi-periodicity) conditions which have to be matched for Fay-like identities at genus zero. The situation is different for matching the poles: despite having no cycles and therefore no integration kernels with a single simple pole, e.g. no one-form in the sense of eq. (2.3), the closest one gets is with kernels

$$\frac{\mathrm{d}z}{z-a} \tag{6.1}$$

that have simple poles at $z = a, \infty$ on the Riemann sphere. As a result, there does not exist a direct relationship wherein Enriquez' connection and the corresponding quadratic identities are reduced to genus zero. Nevertheless, the pole-matching pattern is met by the partial fraction identity

$$\frac{\mathrm{d}z}{z-x}\frac{\mathrm{d}y}{y-x} - \frac{\mathrm{d}z}{z-y}\frac{\mathrm{d}y}{y-x} - \frac{\mathrm{d}z}{z-x}\frac{\mathrm{d}y}{y-z} = 0. \tag{6.2}$$

The cycle structure at genus one can be mathematically realized by averaging over the genus-zero kernels in eq. (6.1). One efficient way of performing the average is to use the Schottky parametrization and has been discussed in [31, Eqn. (5.22)]. A similar approach expressing genus-one polylogarithms in terms of genus-zero differentials has been discussed in ref. [9].

## 6.2 Genus one

As discussed in § 2, the algebra $\mathfrak{b}$ used in Definition 1 will be abelian, that is there is just a single pair of generators $a = a_1$ and $b = b_1$. The meromorphic and quasi-periodic generating function for integration kernels at genus one is the Kronecker function [9],

$$F(\xi - \chi, \alpha) = \sum_{m=0}^{\infty} g^{(m)}(\xi - \chi)\alpha^{m-1}, \tag{6.3}$$

where $\mathfrak{u}$ is the Abel map, $\xi = \mathfrak{u}(z)$ and $\chi = \mathfrak{u}(x)$ are the images of points $z$ and $x$ on the universal cover, and $\alpha$ is the chosen power-counting variable. To compare to Enriquez' connection, we can identify the Kronecker form[10]

$$\mathcal{S}_{(\mathfrak{b})}(z,x) = F(\mathfrak{u}(z) - \mathfrak{u}(x), -b_1/2\pi\mathring{\imath})\mathrm{d}z, \tag{6.4}$$

which can be shown to satisfy

$$\mathcal{S}_{(\mathfrak{b})}(\gamma_1 z, x) = e^{b_1}\mathcal{S}(z,x), \quad \operatorname*{Res}_{z=x}\mathcal{S}_{(\mathfrak{b})}(z,x) = 1. \tag{6.5}$$

One can immediately use the conditions satisfied by Enriquez' connection eq. (2.2) and the first component form eq. (3.5a) at genus one to show

$$K(z,x)\Big|_{h=1} = -\frac{1}{2\pi\mathring{\imath}}\mathcal{S}_{(\mathfrak{b})}(z,x)b_1 a_1, \quad K_{(\mathfrak{b})1}(z,x)\Big|_{h=1} = -\frac{1}{2\pi\mathring{\imath}}\mathcal{S}_{(\mathfrak{b})}(z,x)b_1, \tag{6.6}$$

implying

$$\mathcal{S}_{(\mathfrak{b})}(z,x) = -\frac{1}{2\pi\mathring{\imath}}\sum_{m=0}^{\infty} \omega_{\underbrace{1\ldots 1}_{m \text{ times}}}(z,x)b_1^m. \tag{6.7}$$

---

[10]The convention for the power-counting variables for the Kronecker function and Enriquez' connection differ by a factor of $(-2\pi\mathring{\imath})$. Furthermore, we use the calligraphic $\mathcal{S}$ instead of $S$ as done in ref. [31] to avoid confusion with the antipode.

**Corollary 13.** *The quadratic Fay-like identity for the component forms eq. (4.11),*

$$
\begin{aligned}
0 = & \left[ K_{(\mathfrak{b}_1)l}(z,x) - K_{(\mathfrak{b}_1)l}(z,y) \right] K_{(\mathfrak{b}_2)k}(y,x)(1 \otimes b_m) \\
& + \left[ K_{(\mathfrak{b}_2)m}(y,x) - K_{(\mathfrak{b}_2)m}(y,z) \right] K_{(\mathfrak{b}_1)k}(z,x)(b_l \otimes 1) \\
& - \left( K_{(\mathfrak{b}_1)j}(z,y) K_{(\mathfrak{b}_2)j(\Delta\mathfrak{b}_{12})k}(y,x) + K_{(\mathfrak{b}_2)j}(y,z) K_{(\mathfrak{b}_1)j(\Delta\mathfrak{b}_{12})k}(z,x) \right)(b_l \otimes b_m),
\end{aligned}
\tag{6.8}
$$

*reduces to the Fay identity for the Kronecker function at genus one,*

$$
0 = F(\xi - \chi, \alpha) F(\tilde{\xi} - \chi, \tilde{\alpha}) - F(\xi - \tilde{\xi}, \alpha) F(\tilde{\xi} - \chi, \alpha + \tilde{\alpha}) - F(\xi - \chi, \alpha + \tilde{\alpha}) F(\tilde{\xi} - \xi, \tilde{\alpha})
\tag{6.9}
$$

*for points $\xi, \tilde{\xi}, \chi$ on the universal cover and with formal identification $\alpha = -(b_1 \otimes 1)/(2\pi i)$, $\tilde{\alpha} = -(1 \otimes b_1)/(2\pi i)$.*

*Proof.* Setting $\mathfrak{u}(z) = \xi, \mathfrak{u}(y) = \tilde{\xi}, \mathfrak{u}(x) = \chi$, and multiplying the Fay identity (6.9) by $(b_1 \otimes b_1)\Delta(b_1) \, dz \, dy$, we find that it is equivalent to the Fay identity for the Kronecker form

$$
\begin{aligned}
0 = & \left( \mathcal{S}_{(\mathfrak{b})}(z,x) b_1 \otimes \mathcal{S}_{(\mathfrak{b})}(y,x) b_1 \right) \Delta(b_1) \\
& - \left( \mathcal{S}_{(\mathfrak{b})}(z,y) b_1 \otimes 1 \right) \mathcal{S}_{(\Delta\mathfrak{b}_{12})}(y,x) \Delta(b_1)(1 \otimes b_1) \\
& - \mathcal{S}_{(\Delta\mathfrak{b}_{12})}(z,x) \Delta(b_1)(1 \otimes \mathcal{S}_{(\mathfrak{b})}(y,z) b_1)(b_1 \otimes 1).
\end{aligned}
\tag{6.10}
$$

by using eq. (6.4). On the other hand, one can see that

$$
\begin{aligned}
0 = & K_{(\mathfrak{b}_1)1}(z,x) K_{(\mathfrak{b}_2)1}(y,x)(1 \otimes b_1) + K_{(\mathfrak{b}_1)1}(z,x) K_{(\mathfrak{b}_2)1}(y,x)(b_1 \otimes 1) \\
& - K_{(\mathfrak{b}_1)1}(z,y)[K_{(\mathfrak{b}_2)1}(y,x) + K_{(\mathfrak{b}_2)1(\Delta\mathfrak{b}_{12})1}(y,x)(b_1 \otimes 1)](1 \otimes b_1) \\
& - K_{(\mathfrak{b}_2)1}(y,z)[K_{(\mathfrak{b}_1)1}(z,x) + K_{(\mathfrak{b}_1)1(\Delta\mathfrak{b}_{12})1}(z,x)(1 \otimes b_1)](b_1 \otimes 1)
\end{aligned}
\tag{6.11}
$$

by directly setting all indices to 1 in eq. (4.11). Using the contraction identity eq. (3.17a) (recall that at $h = 1$ algebra $\mathfrak{b}$ is abelian) we find

$$
\begin{aligned}
0 = & K_{(\mathfrak{b}_1)1}(z,x) K_{(\mathfrak{b}_2)1}(y,x)(1 \otimes b_1 + b_1 \otimes 1) \\
& - K_{(\mathfrak{b}_1)1}(z,y) K_{(\Delta\mathfrak{b}_{12})1}(y,x)(1 \otimes b_1) \\
& - K_{(\Delta\mathfrak{b}_{12})1}(z,x) K_{(\mathfrak{b}_2)1}(y,z)(b_1 \otimes 1).
\end{aligned}
\tag{6.12}
$$

This precisely matches the Fay identity for the Kronecker form (6.10) by eq. (6.6) (up to a factor $-1/2\pi i$) and thus shows that eq. (4.11) reduces to the Fay identity for the Kronecker function (6.9). $\qquad \square$

# 7    Application to polylogarithms

Relations between integration kernels of linear, derivative, and quadratic type as in eqs. (2.7), (4.1) and (5.1), respectively, can be used to derive functional relations between higher-genus polylogarithms defined in eq. (2.6). While ref. [45] explored some relations involving Fay-like identities recently, we will consider relations that can involve the appearance of special values, i.e. multiple zeta values here. In the future, these relations can be used to derive identities between higher-genus multiple zeta values.

## 7.1    Removing $z$ from the labels

If we have a higher-genus polylogarithm of the form $\tilde{\Gamma}\left(\begin{smallmatrix} \ldots \ddot{z} \ldots \end{smallmatrix}; z\right)$, where the argument $z$ of the polylogarithm also appears as a label for the pole locations, it can be desired to remove $z$

from the labels. This was already done in the elliptic case [5, Sec. 2.2.2], yielding a functional relation where on the r.h.s. there is an elliptic polylogarithm where $z$ is also a label, while on the l.h.s. only elliptic polylogarithms without $z$ as a label appear and we furthermore can have genus-zero multiple zeta values (MZVs). The same technique can also be applied in the higher-genus case to remove $z$ from the label by using quadratic identities for the kernels.

One obvious question then is whether elliptic zeta values appear appear in these higher-genus polylogarithmic relations, or only MZVs. As we will answer below, the identities will only involve genus-zero MZVs but no elliptic or higher-genus special values.

For the derivation of these functional relations, we start off with a higher-genus polylogarithm

$$\tilde{\Gamma}\left( \begin{smallmatrix} \mathbf{i}_{n_1}, ..., \mathbf{i}_{n_{\ell-1}}, (i_1 \cdots i_r pq), \mathbf{i}_{n_{\ell+1}}, ..., \mathbf{i}_{n_k} \\ x_1, ..., x_{\ell-1}, \quad z, \quad x_{\ell+1}, ..., x_k \end{smallmatrix} ; z \right) \tag{7.1}$$

that has the argument $z$ also a label for a pole (at a position $\ell \in \{1, \ldots, k\}$). We are particularly interested in the case $p = q$, since if the two last indices of a kernel do not equal each other, it is independent of the second variable (see eq. (2.7a)), so if $p \neq q$, the variable $z$ in the label of the kernel is irrelevant. Example 15 will look at one of these trivial examples with $p \neq q$, while in Example 14 a more interesting non-trivial identity for the case $p = q$ will be shown.

Following the derivations for the elliptic case from ref. [5], we start off by writing

$$\tilde{\Gamma}\left( \begin{smallmatrix} \mathbf{i}_{n_1}, ..., \mathbf{i}_{n_{\ell-1}}, (i_1 \cdots i_r pq), \mathbf{i}_{n_{\ell+1}}, ..., \mathbf{i}_{n_k} \\ x_1, ..., x_{\ell-1}, \quad z, \quad x_{\ell+1}, ..., x_k \end{smallmatrix} ; z \right) = \int_{t_0=z_0}^{z} dt_0 \underbrace{\frac{d}{dt_0} \tilde{\Gamma}\left( \begin{smallmatrix} \mathbf{i}_{n_1}, ..., \mathbf{i}_{n_{\ell-1}}, (i_1 \cdots i_r pq), \mathbf{i}_{n_{\ell+1}}, ..., \mathbf{i}_{n_k} \\ x_1, ..., x_{\ell-1}, \quad t_0, \quad x_{\ell+1}, ..., x_k \end{smallmatrix} ; t_0 \right)}_{= d_{t_0}}$$

$$+ \lim_{t \to z_0} \tilde{\Gamma}\left( \begin{smallmatrix} \mathbf{i}_{n_1}, ..., \mathbf{i}_{n_{\ell-1}}, (i_1 \cdots i_r pq), \mathbf{i}_{n_{\ell+1}}, ..., \mathbf{i}_{n_k} \\ x_1, ..., x_{\ell-1}, \quad t, \quad x_{\ell+1}, ..., x_k \end{smallmatrix} ; t \right) \tag{7.2}$$

and investigate the two appearing terms separately:

**Derivative term.** Taking the derivative w.r.t. $t_0$ of the higher-genus polylogarithm inside the integral with $1 < \ell < k$ yields

$$\int_{t_0=z_0}^{z} d_{t_0} \tilde{\Gamma}\left( \begin{smallmatrix} \mathbf{i}_{n_1}, ..., \mathbf{i}_{n_{\ell-1}}, (i_1 \cdots i_r pq), \mathbf{i}_{n_{\ell+1}}, ..., \mathbf{i}_{n_k} \\ x_1, ..., x_{\ell-1}, \quad t_0, \quad x_{\ell+1}, ..., x_k \end{smallmatrix} ; t_0 \right)$$

$$= \int_{t_0=z_0}^{z} \omega_{\mathbf{i}_{n_1}}(t_0, x_1) \tilde{\Gamma}\left( \begin{smallmatrix} \mathbf{i}_{n_2}, ..., \mathbf{i}_{n_{\ell-1}}, (i_1 \cdots i_r pq), \mathbf{i}_{n_{\ell+1}}, ..., \mathbf{i}_{n_k} \\ x_2, ..., x_{\ell-1}, \quad t_0, \quad x_{\ell+1}, ..., x_k \end{smallmatrix} ; t_0 \right)$$

$$+ \int_{t_0=z_0}^{z} \left( \prod_{j=1}^{\ell-1} \int_{t_j=z_0}^{t_{j-1}} \omega_{\mathbf{i}_{n_j}}(t_j, x_j) \right) \int_{t_\ell=z_0}^{t_{\ell-1}} \underbrace{d_{t_0} \omega_{i_1 \cdots i_r pq}(t_\ell, t_0)}_{(-1)^r d_{t_\ell} \omega_{i_r \cdots i_1 pq}(t_0, t_\ell)} \tilde{\Gamma}\left( \begin{smallmatrix} \mathbf{i}_{n_{\ell+1}}, ..., \mathbf{i}_{n_k} \\ x_{\ell+1}, ..., x_k \end{smallmatrix} ; t_\ell \right)$$

$$\stackrel{\text{(P.I.)}}{=} \int_{t_0=z_0}^{z} \omega_{\mathbf{i}_{n_1}}(t_0, x_1) \tilde{\Gamma}\left( \begin{smallmatrix} \mathbf{i}_{n_2}, ..., \mathbf{i}_{n_{\ell-1}}, (i_1 \cdots i_r pq), \mathbf{i}_{n_{\ell+1}}, ..., \mathbf{i}_{n_k} \\ x_2, ..., x_{\ell-1}, \quad t_0, \quad x_{\ell+1}, ..., x_k \end{smallmatrix} ; t_0 \right)$$

$$+ (-1)^r \int_{t_0=z_0}^{z} \left( \prod_{j=1}^{\ell-1} \int_{t_j=z_0}^{t_{j-1}} \omega_{\mathbf{i}_{n_j}}(t_j, x_j) \right) \int_{t_\ell=z_0}^{t_{\ell-1}} \left[ d_{t_\ell} \left( \omega_{i_r \cdots i_1 pq}(t_0, t_\ell) \tilde{\Gamma}\left( \begin{smallmatrix} \mathbf{i}_{n_{\ell+1}}, ..., \mathbf{i}_{n_k} \\ x_{\ell+1}, ..., x_k \end{smallmatrix} ; t_\ell \right) \right) \right.$$

$$\left. - \omega_{i_r \cdots i_1 pq}(t_0, t_\ell) \underbrace{d_{t_\ell} \tilde{\Gamma}\left( \begin{smallmatrix} \mathbf{i}_{n_{\ell+1}}, ..., \mathbf{i}_{n_k} \\ x_{\ell+1}, ..., x_k \end{smallmatrix} ; t_\ell \right)}_{\omega_{\mathbf{i}_{n_{\ell+1}}}(t_\ell, x_{\ell+1}) \tilde{\Gamma}\left( \begin{smallmatrix} \mathbf{i}_{n_{\ell+2}}, ..., \mathbf{i}_{n_k} \\ x_{\ell+2}, ..., x_k \end{smallmatrix} ; t_\ell \right)} \right]$$

$$= \int_{t_0=z_0}^{z} \omega_{\mathbf{i}_{n_1}}(t_0, x_1) \tilde{\Gamma}\left( \begin{smallmatrix} \mathbf{i}_{n_2}, ..., \mathbf{i}_{n_{\ell-1}}, (i_1 \cdots i_r pq), \mathbf{i}_{n_{\ell+1}}, ..., \mathbf{i}_{n_k} \\ x_2, ..., x_{\ell-1}, \quad t_0, \quad x_{\ell+1}, ..., x_k \end{smallmatrix} ; t_0 \right)$$

$$+ (-1)^r \int_{t_0=z_0}^{z} \left( \prod_{j=1}^{\ell-2} \int_{t_j=z_0}^{t_{j-1}} \omega_{\mathbf{i}_{n_j}}(t_j, x_j) \right) \int_{t_{\ell-1}=z_0}^{t_{\ell-2}} \omega_{\mathbf{i}_{n_{\ell-1}}}(t_{\ell-1}, x_{\ell-1}) \omega_{i_r \cdots i_1 pq}(t_0, t_{\ell-1}) \tilde{\Gamma}\left( \begin{smallmatrix} \mathbf{i}_{n_{\ell+1}}, ..., \mathbf{i}_{n_k} \\ x_{\ell+1}, ..., x_k \end{smallmatrix} ; t_{\ell-1} \right)$$

$$-(-1)^r \int_{t_0=z_0}^{z} \left(\prod_{j=1}^{\ell-1} \int_{t_j=z_0}^{t_{j-1}} \omega_{\mathbf{i}_{n_j}}(t_j,x_j)\right) \int_{t_\ell=z_0}^{t_{\ell-1}} \omega_{i_r\cdots i_1 pq}(t_0,t_\ell)\,\omega_{\mathbf{i}_{n_{\ell+1}}}(t_\ell,x_{\ell+1})\,\tilde{\Gamma}\left(\begin{smallmatrix} \mathbf{i}_{n_{\ell+2}},\,\ldots,\,\mathbf{i}_{n_k} \\ x_{\ell+2},\,\ldots,\,x_k \end{smallmatrix}; t_\ell\right), \qquad (7.3)$$

where we used identity (4.1) to swap the arguments of the kernel with a derivative acting on it, as well as partial integration (P.I.). Now we encounter quadratic kernel terms inside our iterated integral, which calls for the use of the quadratic 3-point relations derived in §5, to reduce the quadratic terms in such a way, that we have a proper iterated integral again (with only one kernel used per integration step). Thus, application of fitting instances of the quadratic identity eq. (5.1) will reduce the r.h.s. of the equation to iterated integrals, where $z$ is removed from the labels. As the result becomes very lengthy and involved in the general case, we will not state it here but will look at specific examples of this identity later.

If the argument appears to be the pole of the innermost integral ($\ell = k$) the above result simplifies and we find

$$\int_{t_0=z_0}^{z} \mathrm{d}_{t_0}\,\tilde{\Gamma}\left(\begin{smallmatrix} \mathbf{i}_{n_1},\,\ldots,\,\mathbf{i}_{n_{k-1}},\,(i_1\cdots i_r pq) \\ x_1,\,\ldots,\,x_{k-1},\quad\quad t_0 \end{smallmatrix}; t_0\right)$$

$$= \int_{t_0=z_0}^{z} \omega_{\mathbf{i}_{n_1}}(t_0,x_1)\,\tilde{\Gamma}\left(\begin{smallmatrix} \mathbf{i}_{n_2},\,\ldots,\,\mathbf{i}_{n_{k-1}},\,(i_1\cdots i_r pq) \\ x_2,\,\ldots,\,x_{k-1},\quad\quad t_0 \end{smallmatrix}; t_0\right)$$

$$+ \int_{t_0=z_0}^{z} \left(\prod_{j=1}^{k-1} \int_{t_k=z_0}^{t_{j-1}} \omega_{\mathbf{i}_{n_j}}(t_j,x_j)\right) \int_{t_k=z_0}^{t_{k-1}} \underbrace{\mathrm{d}_{t_0}\omega_{i_1\cdots i_r pq}(t_k,t_0)}_{(-1)^r\mathrm{d}_{t_k}\omega_{i_r\cdots i_1 pq}(t_0,t_k)}$$

$$= \int_{z_0}^{z} \omega_{\mathbf{i}_{n_1}}(t_0,x_1)\,\tilde{\Gamma}\left(\begin{smallmatrix} \mathbf{i}_{n_2},\,\ldots,\,\mathbf{i}_{n_{k-1}},\,(i_1\cdots i_r pq) \\ x_2,\,\ldots,\,x_{k-1},\quad\quad t_0 \end{smallmatrix}; t_0\right)$$

$$+ (-1)^r \int_{t_0=z_0}^{z} \left(\prod_{j=1}^{k-1} \int_{t_j=z_0}^{t_{j-1}} \omega_{\mathbf{i}_{n_j}}(t_j,x_j)\right) (\omega_{i_r\cdots i_1 pq}(t_0,t_{k-1}) - \omega_{i_r\cdots i_1 pq}(t_0,z_0))$$

$$= \int_{t_0=z_0}^{z} \omega_{\mathbf{i}_{n_1}}(t_0,x_1)\,\tilde{\Gamma}\left(\begin{smallmatrix} \mathbf{i}_{n_2},\,\ldots,\,\mathbf{i}_{n_{k-1}},\,(i_1\cdots i_r pq) \\ x_2,\,\ldots,\,x_{k-1},\quad\quad t_0 \end{smallmatrix}; t_0\right)$$

$$+ (-1)^r \int_{t_0=z_0}^{z} \left(\prod_{j=1}^{k-2} \int_{t_j=z_0}^{t_{j-1}} \omega_{\mathbf{i}_{n_j}}(t_j,x_j)\right) \int_{t_{k-1}=z_0}^{t_{k-2}} \omega_{\mathbf{i}_{n_{t_{k-1}}}}(t_{k-1},x_{k-1})\,\omega_{i_r\cdots i_1 pq}(t_0,t_{k-1})$$

$$- (-1)^r\,\tilde{\Gamma}\left(\begin{smallmatrix} (i_r\cdots i_1 pq),\,\mathbf{i}_{n_1},\,\ldots,\,\mathbf{i}_{n_{k-1}} \\ z_0,\quad\quad x_1,\,\ldots,\,x_{k-1} \end{smallmatrix}; z\right). \qquad (7.4)$$

The other special case, $\ell = 1$ is more involved: since the outermost integration is divergent in general, a regularization prescription for the higher-genus polylogarithms (and associated higher-genus MZVs) is required. Whenever regularization is required below, we will either comment on how we deal with it or avoid the corresponding polylogarithms. Making regularization rigorous is beyond the scope of the current article and will be part of future work.

A further problem sets the situation at higher genera apart from the elliptic case: denoting by $\tilde{\omega}$ the function part of integration kernels, e.g.

$$\omega_{i_1\cdots i_r jk}(z,x) =: \tilde{\omega}_{i_1\cdots i_r jk}(z,x)\mathrm{d}z, \qquad (7.5)$$

the anti-symmetry in the elliptic case [5]

$$\omega_{\underbrace{1\ldots 1}_{n+1}}(z,x)\Big|_{h=1} = (-1)^n \omega_{\underbrace{1\ldots 1}_{n+1}}(x,z)\Big|_{h=1} \qquad (7.6)$$

could be used to cancel separately divergent terms (as can be seen in the example in eq. (7.7), reduced to $h = 1$). Unfortunately, this anti-symmetry property does not carry over to higher

genera in an obvious way. Even worse, identity (4.1) involves reversal of the order of indices, which prohibits all obvious cancellation identities. The following depth-one example is the higher-genus version of eq. (7.6):

$$
\tilde{\Gamma}\left(\begin{smallmatrix}(i_1\cdots i_r j k)\\ z\end{smallmatrix}; z\right) = \int_{t_0=z_0}^{z} \underbrace{\mathrm{d}t_0 \, \frac{\mathrm{d}}{\mathrm{d}t_0}}_{=:\mathrm{d}_{t_0}} \tilde{\Gamma}\left(\begin{smallmatrix}(i_1\cdots i_r j k)\\ t_0\end{smallmatrix}; t_0\right) + \lim_{t\to z_0} \tilde{\Gamma}\left(\begin{smallmatrix}(i_1\cdots i_r j k)\\ t\end{smallmatrix}; t\right)
$$

$$
= \int_{t_0=z_0}^{z} \lim_{x\to t_0}\left(\mathrm{d}_{t_0}\tilde{\Gamma}\left(\begin{smallmatrix}(i_1\cdots i_r j k)\\ x\end{smallmatrix}; t_0\right) + \mathrm{d}t_0\,\frac{\mathrm{d}}{\mathrm{d}x}\tilde{\Gamma}\left(\begin{smallmatrix}(i_1\cdots i_r j k)\\ x\end{smallmatrix}; t_0\right)\right) + \lim_{t\to z_0}\tilde{\Gamma}\left(\begin{smallmatrix}(i_1\cdots i_r j k)\\ t\end{smallmatrix}; t\right)
$$

$$
= \int_{t_0=z_0}^{z}\lim_{x\to t_0}\left(\omega_{i_1\cdots i_r j k}(t_0,x) + \mathrm{d}t_0 \int_{t_1=z_0}^{t_0}\underbrace{\frac{\mathrm{d}}{\mathrm{d}x}\,\omega_{i_1\cdots i_r j k}(t_1,x)}_{(-1)^r \mathrm{d}t_1 \tilde{\omega}_{i_r\cdots i_1 j k}(x,t_1)}\right)
$$

$$
+ \lim_{t\to z_0}\tilde{\Gamma}\left(\begin{smallmatrix}(i_1\cdots i_r j k)\\ t\end{smallmatrix}; t\right)
$$

$$
= \int_{t_0=z_0}^{z}\lim_{x\to t_0}\mathrm{d}t_0\bigg(\tilde{\omega}_{i_1\cdots i_r j k}(t_0,x) + (-1)^r \left(\tilde{\omega}_{i_r\cdots i_1 j k}(x,t_0) - \tilde{\omega}_{i_r\cdots i_1 j k}(x,z_0)\right)\bigg)
$$

$$
+ \lim_{t\to z_0}\tilde{\Gamma}\left(\begin{smallmatrix}(i_1\cdots i_r j k)\\ t\end{smallmatrix}; t\right), \tag{7.7}
$$

where we used the differential identity (4.1) moving to the fourth line. Taking the above expression as starting point, it would be desirable to make the expected cancellation obvious. Alternatively, one could try to transform rewrite the expression back into polylogarithms, for which a relation between $\tilde{\omega}_{i_1\cdots i_r j k}(z,x)$ and $\tilde{\omega}_{i_r\cdots i_1 j k}(x,z)$ would be useful. However, as alluded to above, such relations are not known to us beyond genus one. Nevertheless, we are going to consider eq. (7.7) for $r=0$ and $r=1$ in Example 16.

**Boundary term.** Let us now consider the boundary term

$$
\lim_{t\to z_0}\tilde{\Gamma}\left(\begin{smallmatrix}\mathbf{i}_{n_1},\,...,\,\mathbf{i}_{n_{\ell-1}},\,\mathbf{i}_{n_\ell},\,\mathbf{i}_{n_{\ell+1}},\,...,\,\mathbf{i}_{n_k}\\ x_1,\,...,\,x_{\ell-1},\,t,\,\,x_{\ell+1},\,...,\,x_k\end{smallmatrix}; t\right). \tag{7.8}
$$

In most cases this term will vanish, because the length of the integration path shrinks to zero as $t\to z_0$. The only non-vanishing situation occurs, if *all* exhibit poles. As discussed around eq. (2.5), the only integration kernels with poles in the fundamental domain are those of the form $\omega_{jj}(z,x)$ for some $j\in\{1,\ldots,h\}$, where the pole is at $z=x$. Lemma 8 of ref. [26] implies the following expansion (no sum over $j$)

$$
\omega_{jj}(z,x) = -\frac{1}{2\pi i}\frac{1}{z-x} + \mathcal{O}(1), \quad \text{for } z\to x, \quad \forall j\in\{1,\ldots,h\}. \tag{7.9}
$$

Using the above expansion, the boundary term for labels $\mathbf{i}_{n_m}=(j_m,j_m)$, for $j_m\in\{1,\ldots,h\}$, can be written as (no implicit summation)

$$
\lim_{t_0\to z_0}\tilde{\Gamma}\left(\begin{smallmatrix}(j_1,j_1),\,...,\,(j_{\ell-1},j_{\ell-1}),\,(j_\ell,j_\ell),\,(j_{\ell+1},j_{\ell+1}),\,...,\,(j_k,j_k)\\ x_1,\,\,...,\,\,x_{\ell-1},\,\,t_0,\,\,x_{\ell+1},\,\,...,\,\,x_k\end{smallmatrix}; t_0\right)
$$

$$
= \lim_{t_0\to z_0}\frac{1}{(-2\pi i)^k}\left(\prod_{m=1}^{\ell-1}\int_{z_0}^{t_{m-1}}\frac{\mathrm{d}t_m}{t_m-x_m}\right)\int_{z_0}^{t_{\ell-1}}\frac{\mathrm{d}t_\ell}{t_\ell-t_0}\left(\prod_{m=\ell+1}^{k}\int_{z_0}^{t_{m-1}}\frac{\mathrm{d}t_m}{t_m-x_m}\right). \tag{7.10}
$$

The above expression vanishes unless all $x_m\in\{z_0,t_0\}$ as visible after applying substitution (7.12) below.

If all $x_m \in \{z_0, t_0\}$, we find (using $a_m \in \{0,1\}$ with $a_m = 0$ if $x_m = z_0$ and $a_m = 1$ if $x_m = t_0$)

$$
\begin{aligned}
\lim_{t_0 \to z_0} \tilde{\Gamma} \left( \begin{smallmatrix} (j_1,j_1), & ..., & (j_k,j_k) \\ x_1, & ..., & x_k \end{smallmatrix} ; t_0 \right) &= \lim_{t_0 \to z_0} \tilde{\Gamma} \left( \begin{smallmatrix} (j_1,j_1), & ..., & (j_k,j_k) \\ a_1(t_0-z_0)+z_0, & ..., & a_k(t_0-z_0)+z_0 \end{smallmatrix} ; t_0 \right) \\
&= \lim_{t_0 \to z_0} \frac{1}{(-2\pi i)^k} \prod_{m=1}^{k} \int_{z_0}^{t_{m-1}} \frac{dt_m}{t_m - a_m(t_0 - z_0) - z_0} \\
&= \lim_{t_0 \to z_0} \frac{1}{(-2\pi i)^k} \prod_{m=1}^{k} \int_{0}^{t'_{m-1}} \frac{dt'_m}{t'_m - a_m} \\
&= \frac{1}{(-2\pi i)^k} G(a_1, \ldots, a_k; 1),
\end{aligned}
\tag{7.11}
$$

where we used the substitution

$$
t'_i = \frac{t_i - z_0}{t_0 - z_0}, \quad i = 1, \ldots, k,
\tag{7.12}
$$

with $t'_0 = 1$ and where $G(\cdots, z)$ is the Goncharov polylogarithm [17]. When evaluated at $z = 1$, Goncharov polylogarithms constitute MZVs via:

$$
G(\underbrace{0, \ldots, 0}_{n_r - 1}, 1, \underbrace{0, \ldots, 0}_{n_{r-1} - 1}, 1, \ldots, \underbrace{0, \ldots, 0}_{n_1 - 1}, 1; 1) = (-1)^r \zeta(n_1, \ldots, n_r).
\tag{7.13}
$$

In summary, the boundary value (7.11) vanishes in most cases and otherwise yields (up to factors of $-2\pi i$) a genus-zero MZV. If polylogarithms are considered which need regularization, this MZV from the boundary might also require to be regularized, which can be implemented through the usual shuffle regularization of MZVs.

Now that we have investigated the general cases of these higher-genus polylogarithms identities, we will exemplify this in the following, where specifically Example 14 is remarkable due to the appearance of MZVs. Note, that in all of these examples there is no implicit summation over repeated indices.

**Example 14.** Let $j, k \in \{1, \ldots, h\}$, then as in eq. (7.2), we can write (no summation over the indices $j, k$)

$$
\tilde{\Gamma} \left( \begin{smallmatrix} (jj), (kk) \\ z_0, \ z \end{smallmatrix} ; z \right) = \int_{t_0=z_0}^{z} d_{t_0} \tilde{\Gamma} \left( \begin{smallmatrix} (jj), (kk) \\ z_0, \ t_0 \end{smallmatrix} ; t_0 \right) + \lim_{t \to z_0} \tilde{\Gamma} \left( \begin{smallmatrix} (jj), (kk) \\ z_0, \ t \end{smallmatrix} ; t \right).
\tag{7.14}
$$

For the derivative term, using the quadratic identity (5.1) on eq. (7.4), we find:

$$
\begin{aligned}
&\int_{z_0}^{z} d_{t_0} \tilde{\Gamma} \left( \begin{smallmatrix} (jj), (kk) \\ z_0, \ t_0 \end{smallmatrix} ; t_0 \right) \\
&= \int_{t_0=z_0}^{z} \omega_{jj}(t_0, z_0) \int_{t_1=z_0}^{t_0} \omega_{kk}(t_1, t_0) - \int_{t_0=z_0}^{z} \omega_{kk}(t_0, z_0) \int_{t_1=z_0}^{t_0} \omega_{jj}(t_1, z_0) \\
&\quad + \int_{t_0=z_0}^{z} \omega_{kk}(t_0, z_0) \int_{t_1=z_0}^{t_0} \omega_{jj}(t_1, z_0) \\
&\quad - \sum_{m=1}^{h} \left[ \int_{t_0=z_0}^{z} \omega_m(t_0) \int_{t_1=z_0}^{t_0} \omega_{mjj}(t_1, z_0) + \int_{t_0=z_0}^{z} \omega_m(t_0) \int_{t_1=z_0}^{t_0} \omega_{jmj}(t_1, t_0) \right. \\
&\quad \left. + \int_{t_0=z_0}^{z} \omega_{mjj}(t_0, z_0) \int_{t_1=z_0}^{t_0} \omega_m(t_1) + \int_{t_0=z_0}^{z} \omega_{mj}(t_0, z_0) \int_{t_1=z_0}^{t_0} \omega_{jm}(t_1, t_0) \right]
\end{aligned}
$$

$$= \tilde{\Gamma} \left( \begin{smallmatrix} (jj), (kk) \\ z_0, \ x_0 \end{smallmatrix} ; z \right)$$

$$- \sum_{m=1}^{h} \left[ \tilde{\Gamma} \left( \begin{smallmatrix} (m), (mjj) \\ -, \ z_0 \end{smallmatrix} ; z \right) + \tilde{\Gamma} \left( \begin{smallmatrix} (m), (jmj) \\ -, \ z_0 \end{smallmatrix} ; z \right) + \tilde{\Gamma} \left( \begin{smallmatrix} (mjj), (m) \\ z_0, \ - \end{smallmatrix} ; z \right) + \tilde{\Gamma} \left( \begin{smallmatrix} (mj), (jm) \\ z_0, \ x_0 \end{smallmatrix} ; z \right) \right], \tag{7.15}$$

where there is no implicit summation over $j$ and $k$, and "$-$" denotes that the corresponding kernel is independent of the second variable and $x_0$ is an arbitrary point on the universal cover. To reduce all integrals to higher-genus polylogarithms in the above equation, we furthermore used the identity (no summation, and arbitrary $j, m$) $\tilde{\Gamma} \left( \begin{smallmatrix} (jmj) \\ t_0 \end{smallmatrix} ; t_0 \right) = \tilde{\Gamma} \left( \begin{smallmatrix} (jmj) \\ z_0 \end{smallmatrix} ; t_0 \right)$, which is an instance of the identity [45, Eqn. (9.51)] and identity (2.7b).

For the boundary value, following the derivation in eq. (7.11), we find an MZV (no summation over $j, k$):

$$\lim_{z \to z_0} \tilde{\Gamma} \left( \begin{smallmatrix} (jj), (kk) \\ z_0, \ z \end{smallmatrix} ; z \right) = \frac{1}{(-2\pi\mathring{\imath})^2} G(0, 1; 1) = - \frac{1}{(-2\pi\mathring{\imath})^2} \zeta(2) = \frac{1}{24}. \tag{7.16}$$

For genus $h = 2$ and $j = k = 1$, identity (7.14) reduces to

$$\tilde{\Gamma} \left( \begin{smallmatrix} (11), (11) \\ z_0, \ z \end{smallmatrix} ; z \right) = - \tilde{\Gamma} \left( \begin{smallmatrix} (111), (1) \\ z_0, \ - \end{smallmatrix} ; z \right) - \tilde{\Gamma} \left( \begin{smallmatrix} (211), (2) \\ z_0, \ - \end{smallmatrix} ; z \right) - \tilde{\Gamma} \left( \begin{smallmatrix} (21), (12) \\ -, \ - \end{smallmatrix} ; z \right)$$

$$- \tilde{\Gamma} \left( \begin{smallmatrix} (2), (121) \\ -, \ - \end{smallmatrix} ; z \right) - \tilde{\Gamma} \left( \begin{smallmatrix} (2), (211) \\ -, \ z_0 \end{smallmatrix} ; z \right) - 2 \tilde{\Gamma} \left( \begin{smallmatrix} (1), (111) \\ -, \ z_0 \end{smallmatrix} ; z \right) + \frac{1}{24}. \tag{7.17}$$

Using the tools presented in ref. [31], identities like this one can also be verified numerically.

**Example 15.** Consider now the trivial case, where the two last indices of the innermost kernel are different. For this, we consider a polylogarithm of length two, beginning with the identity $(i, j, k, \ell \in \{1, \dots, h\}$ with $k \neq \ell)$

$$\tilde{\Gamma} \left( \begin{smallmatrix} (i), (jk\ell) \\ x_0, \ z \end{smallmatrix} ; z \right) = \int_{t_0 = z_0}^{z} \mathrm{d}t_0 \, \tilde{\Gamma} \left( \begin{smallmatrix} (i), (jk\ell) \\ x_0, \ t_0 \end{smallmatrix} ; t_0 \right). \tag{7.18}$$

In the above equation, both kernels do not have a pole and the boundary term vanishes due to the mechanism explained above. The point $x_0$ is arbitrary here (and can even be omitted) since the holomorphic differentials $\omega_i(z, x)$ depend on the first variable $z$ only. Adapted to this scenario eq. (7.4) generates

$$\tilde{\Gamma} \left( \begin{smallmatrix} (i), (jk\ell) \\ x_0, \ z \end{smallmatrix} ; z \right) = \int_{t_0=z_0}^{z} \omega_i(t_0) \int_{t_1=z_0}^{t_0} \omega_{jk\ell}(t_1, t_0) + \int_{t_0=z_0}^{z} \omega_{jk\ell}(t_0, z_0) \int_{t_1=z_0}^{t_0} \omega_i(t_1)$$

$$- \int_{t_0=z_0}^{z} \int_{t_1=z_0}^{t_0} \omega_i(t_1) \, \omega_{jk\ell}(t_0, t_1)$$

$$= \int_{t_0=z_0}^{z} \omega_i(t_0) \int_{t_1=z_0}^{t_0} \omega_{jk\ell}(t_1, t_0) + \int_{t_0=z_0}^{z} \omega_{jk\ell}(t_0, z_0) \int_{t_1=z_0}^{t_0} \omega_i(t_1)$$

$$- \int_{t_0=z_0}^{z} \omega_i(t_0) \int_{t_1=z_0}^{t_0} \omega_{jk\ell}(t_1, t_0) - \int_{t_0=z_0}^{z} \omega_{jk\ell}(t_0, a) \int_{t_1=z_0}^{t_0} \omega_i(t_1)$$

$$+ \int_{t_0=z_0}^{z} \omega_i(t_0) \int_{t_1=z_0}^{t_0} \omega_{jk\ell}(t_1, b)$$

$$= \tilde{\Gamma} \left( \begin{smallmatrix} (jk\ell), (i) \\ z_0, \ x_0 \end{smallmatrix} ; z \right) - \tilde{\Gamma} \left( \begin{smallmatrix} (jk\ell), (i) \\ a, \ x_0 \end{smallmatrix} ; z \right) + \tilde{\Gamma} \left( \begin{smallmatrix} (i), (jk\ell) \\ x_0, \ b \end{smallmatrix} ; z \right), \tag{7.19}$$

where in the next-to-last step we applied the quadratic identity eq. (5.1) to reduce the product $\omega_i(t_1)\,\omega_{jk\ell}(t_0,t_1)$. The points $a$, $b$ can be chosen on the universal cover arbitrarily. A similar derivation for $j,k,\ell \in \{1,\ldots,h\}$ with $k \neq \ell$ yields the shorter identity

$$\tilde{\Gamma}\left(\begin{smallmatrix}(j),\,(k\ell)\\x_0,\ \ z\end{smallmatrix};z\right) = -\tilde{\Gamma}\left(\begin{smallmatrix}(k\ell),\,(j)\\z_0,\ \ x_0\end{smallmatrix};z\right) + \tilde{\Gamma}\left(\begin{smallmatrix}(k\ell),\,(j)\\a,\ \ x_0\end{smallmatrix};z\right) + \tilde{\Gamma}\left(\begin{smallmatrix}(j),\,(k\ell)\\x_0,\ \ b\end{smallmatrix};z\right), \tag{7.20}$$

where as above $x_0$ is an irrelevant point and $a,b$ are arbitrary points.
Note, that both identities eqs. (7.19), (7.20) are trivial upon noticing that any kernel $\omega_i(z,x)$ or $\omega_{i_1\ldots i_r pq}(z,x)$ for $p \neq q$ is independent of the second variable $x$ (cf. eq. (2.7a)).

**Example 16.** Let us consider the depth-one case, where the first integration kernel carries the label $z$. We consider the formula derived in eq. (7.7) for the cases $r = 0$ and $r = 1$ and will only be concerned with this formula for $j = k$, as other cases are trivial. Throughout this example there is no summation over repeated indices.

- **$r = 0$:** In this case, we deal with the kernel $\omega_{jj}(z,x)$, which has a pole at $z = x$. Separating the pole from the rest of the kernel, i.e. writing

$$\tilde{\omega}_{jj}(z,x) = -\frac{1}{2\pi i}\frac{1}{z-x} + \varpi_j(z,x), \tag{7.21}$$

  with $\varpi_j(z,x)$ a function which is regular at $z = x$, we find in eq. (7.7) for the case $r = 0$ the result

$$\tilde{\Gamma}\left(\begin{smallmatrix}(jj)\\z\end{smallmatrix};z\right) = \int_{t_0=z_0}^{z}\lim_{x\to t_0} \mathrm{d}t_0\left(-\frac{1}{2\pi i}\frac{1}{t_0-x} + \varpi_j(t_0,x) - \frac{1}{2\pi i}\frac{1}{x-t_0} + \varpi_j(x,t_0)\right)$$
$$-\int_{t_0=z_0}^{z}\underbrace{\lim_{x\to t_0}\mathrm{d}t_0\,\tilde{\omega}_{jj}(x,z_0)}_{\omega_{jj}(t_0,z_0)} + \underbrace{\lim_{t\to z_0}\tilde{\Gamma}\left(\begin{smallmatrix}(jj)\\t\end{smallmatrix};t\right)}_{-\frac{1}{2\pi i}G(1;1)}$$
$$= \int_{t_0=z_0}^{z}\lim_{x\to t_0}\mathrm{d}t_0\left(\varpi_j(t_0,x) + \varpi_j(x,t_0)\right) - \tilde{\Gamma}\left(\begin{smallmatrix}(jj)\\z_0\end{smallmatrix};z\right) - \frac{1}{2\pi i}G(1;1). \tag{7.22}$$

  There are a couple of remarks necessary for this result: First, for both polylogarithms appearing in this equation, endpoint regularization is required. Secondly, the Goncharov polylogarithm $G(1;1)$ needs to be regularized by setting $G(1;1) \equiv 0$. Lastly, if the function $\varpi(z,x)$ was anti-symmetric under exchange of $z$ and $x$ the integrand in the above equation would vanish, resulting in an instance of the reflection identity [5, Eqn. (2.44)] at genus one. Unfortunately, for genera higher than one, the terms do not cancel to our knowledge. Thus, in the higher-genus case it is not clear to us how to rewrite the above unintegrated expression into higher-genus polylogarithms.

- **$r = 1$:** For this case, the reversal of the labels $i_1\cdots i_r$ is trivial and we encounter a cancellation in the unintegrated term of eq. (7.7) (all functions are regular at $x = t_0$ so that the limit can be taken without problems):

$$\int_{t_0=z_0}^{z}\lim_{x\to t_0}\mathrm{d}t_0\left(\tilde{\omega}_{i_1\ldots i_r jk}(t_0,x) + (-1)^r\left(\tilde{\omega}_{i_r\ldots i_1 jk}(x,t_0) - \tilde{\omega}_{i_r\ldots i_1 jk}(x,z_0)\right)\right)$$
$$\overset{(r=1)}{=} \int_{t_0=z_0}^{z}\lim_{x\to t_0}\mathrm{d}t_0\left(\tilde{\omega}_{ijk}(t_0,x) - \tilde{\omega}_{ijk}(x,t_0) + \tilde{\omega}_{ijk}(x,z_0)\right)$$

$$= \int_{t_0=z_0}^{z} \mathrm{d}t_0 \underbrace{\tilde{\omega}_{ijk}(t_0, z_0)}_{\omega_{ijk}(t_0, z_0)}$$

$$= \tilde{\Gamma}\left(\begin{smallmatrix}(ijk)\\z_0\end{smallmatrix}; z\right). \tag{7.23}$$

Furthermore, the boundary term vanishes and we are left with the identity

$$\tilde{\Gamma}\left(\begin{smallmatrix}(ijk)\\z\end{smallmatrix}; z\right) = \tilde{\Gamma}\left(\begin{smallmatrix}(ijk)\\z_0\end{smallmatrix}; z\right), \tag{7.24}$$

which is an instance of the relation [45, Eqn. (9.51)]. Similar identities arise when all indices $i_1, \ldots, i_r$ are equal and $r$ is odd, as then the reversal of the indices $i_1 \cdots i_r$ is trivial and the $(-1)^r$ factor leads to a cancellation (cf. again [45, Eqn. (9.51)]). However, the general case of arbitrary labels $i_1 \cdots i_r$ does not have such an immediate cancellation.

# 8    Summary and Outlook

In this article we derived Fay-like identities for meromorphic kernels in the construction of polylogarithms on higher-genus Riemann surfaces from relations for Enriquez' meromorphic generating function. Various types of identities can be created by combining generating functions in a way such that residues cancel and periodicity properties are matched. Once considered at genus one, our identities reduce to the well-known Fay identity for the Kronecker function at genus one.

Upon expanding every generating function within the identities into Enriquez' meromorphic integration kernels for polylogarithms, we find all kernel identities conjectured recently in ref. [45]. We prove that the space of identities is exhausted by our identities.

In §7, we provide several examples for functional relations among higher-genus polylogarithms. We can verify the shown relations numerically using the Schottky approach [31]. When specializing to particular arguments of the higher-genus polylogarithms in these functional relations, one can obtain identities for multiple zeta values associated to higher-genus polylogarithms. However, a proper definition of higher-genus multiple zeta values and a survey of their properties is beyond the scope of this article.

There is a couple of immediate open questions resulting from the article:

- For Riemann surfaces of genera zero and one, the Drinfeld [48] and elliptic/KZB associators [49], respectively, have been facilitated for building recursion algorithms for the calculation of various observables in quantum field theory and string theory. Thus, it would be interesting to explore representations for associators for connections on higher-genus surfaces. The most pressing question here is, whether information can be gained from the degeneration of those objects to the known associators at genera zero and one.

- It would be interesting to know to what extent the functional identities implied by Fay-like identities allow to single out bases for cohomologies in classes of higher-genus polylogarithms. This might as well touch upon the question in what way the uniqueness and exhaustiveness of Fay-like identities might help in showing closure under taking primitives.

- Functional relations for higher-genus polylogarithms are believed to imply relations between higher-genus multiple zeta values. Can one extend the collection of examples in

the current article to a survey of relations between higher-genus zeta values leading to a data mine similar to the two existing ones for genus-zero and genus-one multiple zeta values [38, 40]?

- The most pressing question on a structural level is the relation of higher-genus Fay-like identities to the Fay trisecant equation [10, 11].

- In ref. [45] it was conjectured that all identities derived for periodic kernels would carry over to identities for meromorphic kernels. It would be nice to know, whether the opposite direction could be proven: in this case every identity for meromorphic generating functions should have an echo for the properly periodic kernels.

## Acknowledgments

We are grateful to Federico Zerbini and Zhexian Ji for various discussions and work on related projects. We thank Eric D'Hoker and Oliver Schlotterer for comments on a draft version of this article.

The work of K.B., J.B. and E.I. is partially supported by the Swiss National Science Foundation through the NCCR SwissMAP. The research of K.B., J.B. and Y.M. was supported by the Munich Institute for Astro-, Particle and BioPhysics (MIAPbP) which is funded by the Deutsche Forschungsgemeinschaft (DFG, German Research Foundation) under Germany's Excellence Strategy - EXC-2094 - 390783311.

# Appendix

## A Computations with component forms

### A.1 Properties of the component forms

In this appendix, we explicitly show the properties of the component forms (3.3) and (3.7) stated in section §3. Based on the properties (2.4a), (2.4b) and (2.5) of the integration kernels, we can now verify the properties of the component forms directly based of their definitions.

**First component form.** We verify the properties (3.5a), (3.5b) and (3.5c) by direct computation. For the quasi-periodicity in $z$, we obtain

$$
\begin{aligned}
K_{(\flat)j}(\gamma_k z, x) &= \sum_{r \geq 0} \sum_{m=0}^{r} \frac{1}{m!} \delta_{k i_1 \ldots i_m} \omega_{i_{m+1} \ldots i_r j}(z, x) b_{i_1} \ldots b_{i_r} \\
&= \sum_{r \geq 0} \sum_{m=0}^{r} \frac{1}{m!} \omega_{i_{m+1} \ldots i_r j}(z, x) b_k^m b_{i_{m+1}} \ldots b_{i_r} \\
&= \sum_{m \geq 0} \frac{b_k^m}{m!} \sum_{r \geq m} \omega_{i_{m+1} \ldots i_r j}(z, x) b_{i_{m+1}} \ldots b_{i_r} \\
&= e^{b_k} K_{(\flat)j}(z, x) ,
\end{aligned}
\tag{A.1}
$$

where we have used eq. (2.4a) in the first line, resolved the Kronecker delta in the second line, regrouped the sums in the third line and relabeled $r \to r - m$, $i_{m+n} \to i_n$ in the last line. Completely analogous, we obtain for the quasi-periodicity in $x$

$$
\begin{aligned}
K_{(\flat)j}(z, \gamma_k x) &= K_{(\flat)j}(z, x) + \sum_{r \geq 0} \delta_{i_r j} \sum_{m=0}^{r-1} \frac{(-1)^{m+1}}{(m+1)!} \delta_{k i_{r-1} \ldots i_{r-m}} \omega_{i_1 \ldots i_{r-m-1} k}(z, x) b_{i_1} \ldots b_{i_r} \\
&= K_{(\flat)j}(z, x) + \sum_{r \geq 1} \sum_{m=1}^{r} \frac{(-1)^m}{m!} \omega_{i_1 \ldots i_{r-m} k}(z, x) b_{i_1} \ldots b_{i_{r-m}} b_k^{m-1} b_j \\
&= K_{(\flat)j}(z, x) + K_{(\flat)k}(z, x) \left( \sum_{m \geq 1} \frac{(-1)^m}{m!} b_k^{m-1} \right) b_j ,
\end{aligned}
\tag{A.2}
$$

where we have used eq. (2.4b) and similar manipulations as above. Using the exponential series, we can formally identify

$$
\sum_{m \geq 1} \frac{(-1)^m}{m!} b_k^{m-1} = \frac{e^{-b_k} - 1}{b_k} ,
\tag{A.3}
$$

thereby arriving at eq. (3.5b). For the residue, eq. (2.5) directly yields eq. (3.5c) upon resolution of the Kronecker delta.

**Second component form.** For the second component form, we perform a very similar procedure, but now paying attention to the second flavor of power counting variables. Again, using

eq. (2.4a) we find

$$
K_{(\mathfrak{b}_1)i(\mathfrak{b}_2)j}(\gamma_k z, x) = \sum_{r,s \geq 0} \left( \sum_{m=0}^{r} \frac{\omega_{i_{m+1}\ldots i_r i p_1 \ldots p_s j}(z,x)}{m!} b_k^m b_{i_{m+1}} \ldots b_{i_r} \otimes b_{p_1} \ldots b_{p_s} \right)
$$

$$
+ \delta_{ik} \sum_{r,s \geq 0} \left( \sum_{m=0}^{s} \sum_{p_{m+1},\ldots,p_s=1}^{h} \frac{\omega_{p_{m+1}\ldots p_s j}(z,x)}{(r+m+1)!} (b_k)^r \otimes (b_k)^m b_{p_{m+1}} \ldots b_{p_s} \right)
$$

$$
= \sum_{m \geq 0} \frac{(b_k^m \otimes 1)}{m!} \sum_{\substack{s \geq 0 \\ r \geq m}} \sum_{\substack{i_{m+1},\ldots,i_r=1 \\ p_1,\ldots,p_s=1}}^{h} \omega_{i_{m+1}\ldots i_r i p_1 \ldots p_s j}(z,x)\, b_{i_{m+1}} \ldots b_{i_r} \otimes b_{p_1} \ldots b_{p_s}
$$

$$
+ \delta_{ik} \sum_{r,m \geq 0} \frac{(b_k)^r \otimes (b_k)^m}{(r+m+1)!} \sum_{\substack{s \geq m \\ p_{m+1},\ldots,p_s=1}}^{h} \omega_{p_{m+1}\ldots p_s j}(z,x)(1 \otimes b_{p_{m+1}} \ldots b_{p_s})
$$

$$
= (e^{b_k} \otimes 1) K_{(\mathfrak{b}_1)i(\mathfrak{b}_2)j}(z,x) + \delta_{ik} \left( \sum_{r,m \geq 0} \frac{(b_k)^r \otimes (b_k)^m}{(r+m+1)!} \right) K_{(\mathfrak{b}_2)j}(z,x)
$$

$$\tag{A.4}$$

for the quasi-periodicity in $z$. This is exactly eq. (3.8). Moreover, we can use the exponential and geometric series to formally identify

$$
\sum_{r,m \geq 0} \frac{(b_k)^r \otimes (b_k)^m}{(r+m+1)!} = \sum_{r \geq 0} \left( b_k \otimes b_k^{-1} \right)^r \left( 1 \otimes \frac{e^{b_k}}{b_k} - 1 \otimes \sum_{m=0}^{r} \frac{(b_k)^{m-1}}{m!} \right)
$$

$$
= \frac{1 \otimes e^{b_k}}{1 \otimes b_k - b_k \otimes 1} - \sum_{r \geq 0} \sum_{m \geq 0} \left( b_k \otimes b_k^{-1} \right)^r \frac{1}{1 \otimes b_k} \frac{(b_k)^m \otimes 1}{m!} \tag{A.5}
$$

$$
= \frac{1 \otimes e^{b_k} - e^{b_k} \otimes 1}{1 \otimes b_k - b_k \otimes 1}.
$$

For the quasi-periodicity in $x$ in the special case of $K_{(\mathfrak{b})ij}(z,x)$, we again use eq. (2.4b) to derive

$$
K_{(\mathfrak{b})ij}(z, \gamma_k x) = K_{(\mathfrak{b})ij}(z,x) + \delta_{ij} \sum_{r \geq 0} \sum_{m=0}^{r} \frac{(-1)^{m+1}}{(m+1)!} \sum_{i_1,\ldots,i_{r-m}=1}^{h} \omega_{i_1 \ldots i_{r-m} k}(z,x)\, b_{i_1} \ldots b_{i_{r-m}} b_k^m
$$

$$
= K_{(\mathfrak{b})ij}(z,x) + \delta_{ij} \sum_{r \geq 0} \sum_{i_1,\ldots,i_r=1}^{h} \omega_{i_1 \ldots i_r k}(z,x)\, b_{i_1} \ldots b_{i_r} \left( \sum_{m \geq 1} \frac{(-1)^m}{m!} b_k^{m-1} \right)
$$

$$
= K_{(\mathfrak{b})ij}(z,x) + \delta_{ij} K_{(\mathfrak{b})k}(z,x) \frac{e^{-b_k} - 1}{b_k}, \tag{A.6}
$$

which coincides with eq. (3.12). The residue follows directly from resolution of the Kronecker delta in this case as well. We can also do this for the variations of the component forms involving

the Hopf algebra morphisms. For example, we have

$$
\begin{aligned}
K_{(\mathfrak{b}_2)i(\Delta\mathfrak{b}_{12})j}(\gamma_k z, x) &= \mu_{23} \circ \mathcal{P}_{12} \circ \Delta_2(K_{(\mathfrak{b}_1)i(\mathfrak{b}_2)j}(\gamma_k z, x)) \\
&= \mu_{23} \circ \mathcal{P}_{12} \circ \Delta_2\Bigg( (e^{b_k} \otimes 1) K_{(\mathfrak{b}_1)i(\mathfrak{b}_2)j}(z, x) \\
&\qquad + \delta_{ik}\left( \frac{1 \otimes e^{b_k} - e^{b_k} \otimes 1}{1 \otimes b_k - b_k \otimes 1} \right) K_{(\mathfrak{b}_2)j}(z, x) \Bigg) \\
&= \mu_{23} \circ \mathcal{P}_{12}\Bigg( (e^{b_k} \otimes 1 \otimes 1) K_{(\mathfrak{b}_1)i(\Delta\mathfrak{b}_{23})j}(z, x) \\
&\qquad + \delta_{ik}\left( \frac{1 \otimes e^{b_k} \otimes e^{b_k} - e^{b_k} \otimes 1 \otimes 1}{1 \otimes 1 \otimes b_k + 1 \otimes b_k \otimes 1 - b_k \otimes 1 \otimes 1} \right) K_{(\Delta\mathfrak{b}_{23})j}(z, x) \Bigg) \\
&= (1 \otimes e^{b_k}) K_{(\mathfrak{b}_2)i(\Delta\mathfrak{b}_{12})j}(z, x) + \delta_{ik} \frac{e^{b_k} \otimes e^{b_k} - 1 \otimes e^{b_k}}{b_k \otimes 1} K_{(\Delta\mathfrak{b}_{12})j}(z, x),
\end{aligned}
\tag{A.7}
$$

where we have used that the Hopf algebra maps are homomorphisms. Analogously, the quasi-periodicity of arbitrary variations of the component forms can be computed.

## A.2 Properties of the quadratic identity generating function

In this appendix we derive the quasi-periodicity property as well as the residues of the generating function for the Fay-like identities defined in eq. (4.11). To show the analytical properties of $\mathcal{Q}_{klm}(z, y, x)$, we can focus on $q_{klm}(z, y, x)$. We start by deriving the quasi-periodicity in $z$. By using eq. (3.5a), we immediately obtain

$$
q_{klm}(\gamma_i z, y, x) = (e^{b_i} \otimes 1) q_{klm}(z, y, x).
\tag{A.8}
$$

The quasi-periodicity in $y$ is a little bit more involved, but applying equations (3.5a), (3.5b) and (3.8) and using the result (A.7) allows us to formally deduce

$$
\begin{aligned}
q_{klm}(z, \gamma_i y, x) &= (1 \otimes e^{b_i}) \left[ K_{(\mathfrak{b}_1)l}(z, x) - K_{(\mathfrak{b}_1)l}(z, y) \right] K_{(\mathfrak{b}_2)k}(y, x)(1 \otimes b_m) \\
&\quad - (1 \otimes e^{b_i}) K_{(\mathfrak{b}_1)j}(z, y) K_{(\mathfrak{b}_2)j(\Delta\mathfrak{b}_{12})k}(y, x)(b_l \otimes b_m) \\
&\quad - \Bigg[ \delta_{ij} K_{(\mathfrak{b}_1)j}(z, y) \frac{e^{\Delta b_i} - 1 \otimes e^{b_i}}{b_i \otimes 1} K_{(\Delta\mathfrak{b}_{12})k}(y, x) \\
&\qquad + K_{(\mathfrak{b}_1)i}(z, y) \frac{e^{-b_i} - 1}{b_i} \otimes e^{b_i} K_{(\mathfrak{b}_2)k}(y, x) \\
&\qquad + K_{(\mathfrak{b}_1)i}(z, y) \frac{e^{-b_i} - 1}{b_i} b_j \otimes e^{b_i} K_{(\mathfrak{b}_2)j(\Delta\mathfrak{b}_{12})k}(y, x) \\
&\qquad + \delta_{ij} K_{(\mathfrak{b}_1)i}(z, y) \left( \frac{e^{-b_i} - 1}{b_i} b_j \otimes 1 \right) \frac{e^{\Delta(b_i)} - 1 \otimes e^{b_i}}{b_i \otimes 1} K_{(\Delta\mathfrak{b}_{12})k}(y, x) \Bigg] (b_l \otimes b_m) \\
&= (1 \otimes e^{b_i}) q_{klm}(z, y, x) \\
&\quad + K_{(\mathfrak{b}_1)i}(z, y) \frac{e^{-b_i} - 1}{b_i} \otimes e^{b_i} \Big[ K_{(\Delta\mathfrak{b}_{12})k}(y, x) - K_{(\mathfrak{b}_2)k}(y, x) \\
&\qquad\qquad\qquad\qquad\qquad - (b_j \otimes 1) K_{(\mathfrak{b}_2)j(\Delta\mathfrak{b}_{12})k}(y, x) \Big] (b_l \otimes b_m) \\
&= (1 \otimes e^{b_i}) q_{klm}(z, y, x)
\end{aligned}
\tag{A.9}
$$

by means of the contraction identity (3.17b). Let us consider the residues next. Using eqs. (3.5c) and (3.9), we arrive at the conditions

$$(-2\pi i) \operatorname*{Res}_{z=x} q_{klm}(z,y,x) = K_{(\mathfrak{b}_2)k}(y,x)\, b_l \otimes b_m, \tag{A.10a}$$

$$(-2\pi i) \operatorname*{Res}_{y=x} q_{klm}(z,y,x) = -K_{(\mathfrak{b}_1)k}(z,x)\, b_l \otimes b_m, \tag{A.10b}$$

$$(-2\pi i) \operatorname*{Res}_{z=y} q_{klm}(z,y,x) = K_{(\Delta\mathfrak{b}_{12})k}(y,x)\, b_l \otimes b_m, \tag{A.10c}$$

where we have again used eq. (3.17b) to derive equation (A.10c). Having established these properties, we can use them to derive the analogous properties of the full expression $\mathcal{Q}_{klm}(z,y,x)$ via eq. (4.11). We get

$$\begin{aligned}
\mathcal{Q}_{klm}(\gamma_l z, y, x) &= q_{klm}(\gamma_l z, y, x) + \mathcal{P}_{12}(q_{kml}(y, \gamma_l z, x)) \\
&= \Big( (e^{b_l} \otimes 1) q_{klm}(z,y,x) + \mathcal{P}_{12}((1 \otimes e^{b_l}) q_{kml}(y,z,x)) \Big) \\
&= (e^{b_l} \otimes 1) \mathcal{Q}_{klm}(z,y,x), \tag{A.11a}
\end{aligned}$$

$$\mathcal{Q}_{klm}(z, \gamma_l y, x) = (1 \otimes e^{b_l}) \mathcal{Q}_{klm}(z,y,x), \tag{A.11b}$$

by means of eqs. (A.8) and (A.9). Moreover, we have

$$\begin{aligned}
(-2\pi i) \operatorname*{Res}_{z=x} \mathcal{Q}_{klm}(z,y,x) &= (-2\pi i) \operatorname*{Res}_{z=x} q_{klm}(z,y,x) + \mathcal{P}_{12} \operatorname*{Res}_{z=x} q_{kml}(y,z,x) \\
&= K_{(\mathfrak{b}_2)k}(y,x) b_l \otimes b_m + \mathcal{P}_{12}\left(-K_{(\mathfrak{b}_1)k}(y,x) b_m \otimes b_l\right) = 0 \tag{A.12a}
\end{aligned}$$

$$(-2\pi i) \operatorname*{Res}_{y=x} \mathcal{Q}_{klm}(z,y,x) = \left(K_{(\mathfrak{b}_1)k}(z,x) - K_{(\mathfrak{b}_1)k}(z,x)\right) b_l \otimes b_m = 0, \tag{A.12b}$$

$$(-2\pi i) \operatorname*{Res}_{z=y} \mathcal{Q}_{klm}(z,y,x) = (-2\pi i) \left[ \operatorname*{Res}_{z=y} q_{klm}(z,y,x) + \operatorname*{Res}_{z=y} \mathcal{P}_{12}(q_{kml}(y,z,x)) \right] = 0, \tag{A.12c}$$

where we have used eqs. (A.10a), (A.10b) and (A.10c). Additionally, eq. (A.12c) requires that

$$\operatorname*{Res}_{z=y} \mathcal{P}_{12}(q_{kml}(y,z,x)) = -\operatorname*{Res}_{z=y} q_{klm}(z,y,x). \tag{A.13}$$

This condition can be reduced to the requirement

$$(-2\pi i) \operatorname*{Res}_{z=y} K_{(\mathfrak{b})k}(y,z) = -b_k. \tag{A.14}$$

To see that this is true, consider the Laurent expansion of $K_{(\mathfrak{b})k}(z,y)$, which reads

$$K_{(\mathfrak{b})k}(z,y) = \frac{b_k}{(-2\pi i)} \frac{1}{z-y} + \mathcal{O}(1). \tag{A.15}$$

From this, we can directly read off that eq. (A.14) is true as exchanging the roles of $z$ and $y$ introduces an additional minus sign.

## A.3 Uniqueness of generating functions valued in the tensor algebra

In the proof of Theorem 9, we identified the one-form $\Omega_{klm} \in \mathfrak{b} \otimes \mathfrak{b}$. This one-form is holomorphic, and satisfies

$$\Omega_{klm}(z, \gamma_i y, x) = (1 \otimes e^{b_i}) \Omega_{klm}(z,y,x). \tag{A.16}$$

By expanding $\Omega_{klm}$ in the first algebra $\mathfrak{b}$, we can identify its components

$$\Omega_{klm}(z,y,x) = \sum_{r=0}^{\infty} b_{i_1} \cdots b_{i_r} \otimes \Omega_{klm,i_1\cdots i_r}(z,y,x), \tag{A.17}$$

where $\Omega_{klm,i_1\cdots i_r}$ are also holomorphic and quasi-periodic,

$$\Omega_{klm,i_1\cdots i_r}(z,\gamma_i y,x) = e^{b_i}\Omega_{klm,i_1\cdots i_r}(z,\gamma_i y,x). \tag{A.18}$$

The proof of Theorem 9 uses the statement that $\Omega_{klm}$ is uniquely identified to vanish by its holomorphicity and quasi-periodicity requirements. We proceed to show this by adapting the theorem used by ref. [27] for identifying the uniqueness of Enriquez' connection.

**Theorem 17** (Uniqueness of algebra-valued one-forms). *A one-form $H(y,x) \in \mathfrak{b}$ can be uniquely identified if it satisfies the conditions*

$$(2\pi\mathring{\imath})\operatorname*{Res}_{y=x} H(y,x) = \sum_{j=1}^{h} b_j r_j, \tag{A.19a}$$

$$H(\gamma_i y, x) = e^{b_i} H(y,x), \tag{A.19b}$$

*for some $r_j \in \mathfrak{b}$. In other words, if $H$ and $H'$ satisfy these conditions for the same $r_j$, then $H \equiv H'$.*

*Proof.* The proof proceeds in two parts. First, we show that the residue and quasi-periodicity conditions imply a condition on the value of $H$'s A-periods. Then, we deduce conditions on the coefficients of $G$ and see that the residue, quasi-periodicity, and A-periods uniquely identify them.

By performing a canonical dissection of the Riemann surface $\Sigma$, we identify the boundary of the fundamental domain with the sequence of cycles $\mathfrak{B}_1\mathfrak{A}_1\mathfrak{B}_1^{-1}\mathfrak{A}_1^{-1}\cdots\mathfrak{B}_h\mathfrak{A}_h\mathfrak{B}_h^{-1}\mathfrak{A}_h^{-1}$. Then, integrating along this boundary we find

$$\oint_{\partial\Sigma} H(y,x) = (2\pi\mathring{\imath})\operatorname*{Res}_{y=x} H(y,x) = \sum_{j=1}^{h} b_j r_j$$

$$= \sum_{j=1}^{h}\oint_{\mathfrak{B}_j\mathfrak{A}_j\mathfrak{B}_j^{-1}\mathfrak{A}_j^{-1}} H(y,x) = \sum_{j=1}^{h}\oint_{\mathfrak{A}_j} H(\gamma_i y,x) - H(y,x) = \sum_{j=1}^{h}(e^{b_i}-1)\oint_{\mathfrak{A}_j} H(y,x). \tag{A.20}$$

By collecting words with the same starting letter, one finds

$$\frac{e^{b_j}-1}{b_j}\oint_{\mathfrak{A}_j} H(y,x) = r_j \implies \oint_{\mathfrak{A}_j} H(y,x) = \frac{b_j}{e^{b_j}-1} r_j, \tag{A.21}$$

where $\frac{b_j}{e^{b_j}-1}$ is identified with the formal sum $\sum_{s=0}^{\infty}\frac{B_s(b_j)^s}{s!}$ where $B_s$ are the Bernoulli numbers. Now, by expanding $H(y,x) = \sum_{s=0}^{\infty} h_{i_1\cdots i_s}(y,x)\, b_{i_1}\cdots b_{i_s}$, one identifies the conditions on $h$ as

$$H(\gamma_i y, x) = e^{b_i} H(y,x) \implies h_{i_1\cdots i_s}(\gamma_i y,x) = \sum_{k=0}^{s}\frac{\delta_{i_1\cdots i_k i}}{k!} h_{i_{k+1}\cdots i_s}(y,x),$$

$$(2\pi\mathring{\imath})\operatorname*{Res}_{y=x} H(y,x) = \sum_{j=1}^{h} b_j r_j \implies (2\pi\mathring{\imath})\operatorname*{Res}_{y=x} h_{i_1\cdots i_s}(y,x) = r_{i_1,i_2\cdots i_s}, \tag{A.22}$$

$$\oint_{\mathfrak{A}_j} H(y,x) = \frac{b_j}{e^{b_j}-1} r_j \implies \oint_{\mathfrak{A}_j} h_{i_1\cdots i_s}(y,x) = \sum_{k=0}^{s}\frac{\delta_{i_1\cdots i_k j} B_k}{k!} r_{i_1,i_{k+1}\cdots i_s},$$

where $r_j = \sum_{s=0}^{\infty} r_{j,i_1\cdots i_s} b_{i_1} \cdots b_{i_s}$. These conditions are sufficient to uniquely identify $h_{i_1\cdots i_s}$: if $h$ and $h'$ have the same residue, quasi-periodicity, and A-period, then their difference $h - h'$ is a holomorphic periodic one-form with vanishing A-periods, and can only be identically zero. Since each coefficient of $H$ is uniquely identified, so is $H$ itself. $\qquad\square$

In ref. [27], one can identify $r_j = a_j$ and deduce the uniqueness of Enriquez' connection. In our case, since $\Omega_{klm,i_1\cdots i_r}$ are holomorphic we identify $r_j = 0$. Since 0 satisfies the same quasi-periodicity and residue conditions (i.e. $0 = e^{b_i}0$ and $\mathrm{Res}_{y=x} 0 = 0$), we deduce that $\Omega_{klm,i_1\cdots i_r} \equiv 0$, and consequently that $\Omega_{klm} \equiv 0$.

## A.4 Derivations for Fay-like identities

In this appendix, we will perform several explicit calculations used in §4 and §5. We start by deriving eq. (4.20) from eq. (4.19). First, by applying the index swapping identities eq. (4.4) and eq. (4.7) to the corresponding terms with repeated $z$-dependence. Doing this, we arrive at the expression

$$
\begin{aligned}
\mathcal{Q}'_{klm}(z,y,x) =& \Bigg( \Big[ \big(K_{(\mathfrak{b}_1)pp'}(z,x) - K_{(\mathfrak{b}_1)pp'}(z,y)\big) \Big]\Big|_{p'=p} K_{(\mathfrak{b}_2)k}(y,x) + K_{(\mathfrak{b}_1)k}(z,x)K_{(\mathfrak{b}_2)qq'}(y,x)\Big|_{q'=q} \\
&- K_{(\mathfrak{b}_1)j}(z,y)K_{(\mathfrak{b}_2)j(\Delta\mathfrak{b}_{12})k}(y,x) - K_{(\mathfrak{b}_1)k}(z,x)K_{(\mathfrak{b}_2)qq'}(y,z)\Big|_{q'=q} \\
&- K_{(\mathfrak{b}_2)j}(y,z)K_{(\mathfrak{b}_1)j(\Delta\mathfrak{b}_{12})k}(z,x) \Bigg)(b_l \otimes b_m) \\
=& \Bigg( \Big[ \big(K_{(\mathfrak{b}_1)pp'}(z,x) - K_{(\mathfrak{b}_1)pp'}(z,y)\big) \Big]\Big|_{p'=p} K_{(\mathfrak{b}_2)k}(y,x) + K_{(\mathfrak{b}_1)k}(z,x)K_{(\mathfrak{b}_2)qq'}(y,x)\Big|_{q'=q} \\
&- K_{(\mathfrak{b}_1)j}(z,y)K_{(\mathfrak{b}_2)j(\Delta\mathfrak{b}_{12})k}(y,x) - \Bigg( K_{(\mathfrak{b}_1)j}(z,x)K_{(\mathfrak{b}_2)jk}(y,z) \\
&- K_{(\mathfrak{b}_1)j}(z,x)K_{(\mathfrak{b}_2)jk}(y,x) + K_{(\mathfrak{b}_1)k}(z,x)K_{(\mathfrak{b}_2)qq'}(y,x)\Big|_{q'=q} \Bigg) \\
&- \Bigg( K_{(\mathfrak{b}_2)j}(y,x)K_{(\mathfrak{b}_1)j(\Delta\mathfrak{b}_{12})k}(z,x) - K_{(\mathfrak{b}_1)j}(z,x)\big[K_{(\mathfrak{b}_2)jk}(y,z) - K_{(\mathfrak{b}_2)jk}(y,x)\big] \\
&+ \big[K_{(\mathfrak{b}_2)jk}(y,z) - K_{(\mathfrak{b}_2)jk}(y,x)\big] K_{(\Delta\mathfrak{b}_{12})j}(z,x) \Bigg) \Bigg)(b_l \otimes b_m) \\
=& \Bigg( \big[K_{(\mathfrak{b}_2)jk}(y,x) - K_{(\mathfrak{b}_2)jk}(y,z)\big] K_{(\Delta\mathfrak{b}_{12})j}(z,x) \\
&+ \big[K_{(\mathfrak{b}_1)jk}(z,x) - K_{(\mathfrak{b}_1)jk}(z,y)\big] K_{(\mathfrak{b}_2)j}(y,x) \\
&- K_{(\mathfrak{b}_2)j}(y,x)K_{(\mathfrak{b}_1)j(\Delta\mathfrak{b}_{12})k}(z,x) - K_{(\mathfrak{b}_1)j}(z,y)K_{(\mathfrak{b}_2)j(\Delta\mathfrak{b}_{12})k}(y,x) \Bigg)(b_l \otimes b_m),
\end{aligned}
$$
(A.23)

where we have again used the index swapping identity (4.4) in the last step.

Next, we verify the transformation (4.21). We can do this using the Hopf algebra axioms. For example, for the first term in eq. (4.20) we have

$$
\Xi\big(K_{(\mathfrak{b}_2)jk}K_{(\Delta\mathfrak{b}_{12})j}\big) = \mathrm{id}\otimes\mu \circ \Delta \otimes \mathrm{id} \circ \mathrm{id}\otimes S \circ \Delta(K_{(\mathfrak{b})j})K_{(S\mathfrak{b}_2)jk}\,.
\tag{A.24}
$$

Then notice

$$\mathrm{id}\otimes\mu\circ\Delta\otimes\mathrm{id}\circ\mathrm{id}\otimes S\circ\Delta = \mathrm{id}\otimes\mu\circ\mathrm{id}\otimes\mathrm{id}\otimes S\circ\Delta\otimes\mathrm{id}\circ\Delta$$
$$= \mathrm{id}\otimes\mu\circ\mathrm{id}\otimes\mathrm{id}\otimes S\circ\mathrm{id}\otimes\Delta\circ\Delta$$
$$= \mathrm{id}\otimes(\eta\circ\epsilon)\circ\Delta = \mathrm{id}\otimes 1, \tag{A.25}$$

where in the first equality we notice that $\Delta_1$ and $S_2$ do not interfere with each other, in the second equality we use coassociativity and in the last equality we use antipode compatibility condition. Then $\mathrm{id}\otimes(\eta\circ\epsilon)$ kills all the non-trivial algebra elements in the second tensor site leaving the only term of the coproduct, when the second tensor site is the neutral element $(\mathrm{id}\otimes 1 : \mathfrak{b}\to\mathfrak{b}^{\otimes 2}$ stands for concatenating a neutral element from the right). The second term in eq. (4.20) is treated analogously. The second term in eq. (4.20) is treated analogously. The second line in eq. (4.20) is obviously mapped to the second line of eq. (4.23). The term of the third line in eq. (4.20) can be shown to be mapped to the third line of eq. (4.23) due to

$$\mathrm{id}\otimes\mu\circ\Delta\otimes\mathrm{id}\circ\mathrm{id}\otimes S\circ\mu\otimes\mathrm{id}\circ\mathrm{id}\otimes\Delta = \mathrm{id}\otimes\mu\circ\Delta\otimes\mathrm{id}\circ\mu\otimes\mathrm{id}\circ\mathrm{id}^{\otimes 2}\otimes S\circ\mathrm{id}\otimes\Delta$$
$$= \mathrm{id}\otimes\mu\circ\mu\otimes\mu\otimes\mathrm{id}\circ\mathrm{id}\otimes\mathcal{P}\otimes\mathrm{id}^{\otimes 2}\circ\Delta\otimes\Delta\otimes\mathrm{id}\circ\mathrm{id}^{\otimes 2}\otimes S\circ\mathrm{id}\otimes\Delta$$
$$= \mu\otimes\mu\circ\mathrm{id}^{\otimes 3}\otimes\mu\circ\mathrm{id}\otimes\mathcal{P}\otimes\mathrm{id}^{\otimes 2}\circ\mathrm{id}^{\otimes 4}\otimes S\circ\Delta\otimes\Delta\otimes\mathrm{id}\circ\mathrm{id}\otimes\Delta$$
$$= \mu\otimes\mu\circ\mathrm{id}\otimes\mathcal{P}\otimes\mathrm{id}\circ\mathrm{id}^{\otimes 3}\otimes\mu\circ\mathrm{id}^{\otimes 4}\otimes S\circ\mathrm{id}^{\otimes 3}\otimes\Delta\circ\Delta\otimes\Delta\otimes\Delta$$
$$= \mu\otimes\mu\circ\mathrm{id}\otimes\mathcal{P}\otimes\mathrm{id}\circ\mathrm{id}^{\otimes 3}\otimes(\eta\circ\epsilon)\circ\Delta\otimes\Delta$$
$$= \mu\otimes\mu\circ\mathrm{id}\otimes\mathcal{P}\otimes\mathrm{id}\circ\Delta\otimes\mathrm{id}\otimes 1$$
$$= \mu\otimes\mathrm{id}\circ\mathrm{id}\otimes\mathcal{P}\circ\Delta\otimes\mathrm{id} = \mu_{12}\circ\mathcal{P}_{23}\circ\Delta_1, \tag{A.26}$$

where we used for each equality: commutativity of $\Delta_1\circ S_2$; compatibility $\mu$ and $\delta$; associativity of $\mu$ and commutativity of $\Delta_1\Delta_2 S_3$; commutativity of $\mu_{45}$ and $\mathcal{P}_{23}$; compatibility of the antipode; property of $\mathrm{id}\otimes(\eta\circ\epsilon)$ when acted on $\Delta$; property $\mu(\cdot\otimes 1)=\mathrm{id}(\cdot)$. The last term in eq. (4.20) can be treated analogously. We have (omitting arguments for brevity)

$$\Xi\left(K_{(\mathfrak{b}_1)j}K_{(\mathfrak{b}_2)j(\Delta\mathfrak{b}_{12})k}\right) = K_{(\Delta\mathfrak{b}_{12})j}\mu_{23}\circ\Delta_1\circ S_2\circ\mu_{23}\circ\mathcal{P}_{12}\circ\Delta_2(K_{(\mathfrak{b}_1)j(\mathfrak{b}_2)k}). \tag{A.27}$$

It can be seen that this is mapped to the last line of eq. (4.23) by

$$\mu_{23}\circ\Delta_1\circ S_2\circ\mu_{23}\circ\mathcal{P}_{12}\circ\Delta_2$$
$$= (\mathrm{id}\otimes\mu)\circ(\Delta\otimes\mathrm{id})\circ(\mathrm{id}\otimes\mu)\circ(\mathrm{id}\otimes\mathcal{P})\circ(\mathrm{id}\otimes S\otimes S)\circ(\mathcal{P}\otimes\mathrm{id})\circ(\mathrm{id}\otimes\Delta)$$
$$= (\mathrm{id}\otimes\mu)\circ(\Delta\otimes\mathrm{id})\circ(\mathrm{id}\otimes\mu)\circ(\mathrm{id}\otimes\mathcal{P})\circ(\mathcal{P}\otimes\mathrm{id})\circ(S\otimes\mathrm{id}\otimes S)\circ(\mathrm{id}\otimes\Delta)$$
$$= (\mathrm{id}\otimes\mu)\circ(\mathrm{id}^{\otimes 2}\otimes\mu)\circ(\Delta\otimes\mathrm{id})\circ(\mathrm{id}\otimes\mathcal{P})\circ(\mathcal{P}\otimes\mathrm{id})\circ(S\otimes\mathrm{id}\otimes S)\circ(\mathrm{id}\otimes\Delta)$$
$$= (\mathrm{id}\otimes\mu)\circ(\mathrm{id}^{\otimes 2}\otimes\mu)\circ(\mathrm{id}^{\otimes 2}\otimes\mathcal{P})\circ(\mathrm{id}\otimes\mathcal{P}\otimes\mathrm{id})\circ(\mathcal{P}\otimes\mathrm{id}^{\otimes 2})\circ(\mathrm{id}\otimes\Delta\otimes\mathrm{id})\circ(S\otimes\mathrm{id}\otimes S)\circ(\mathrm{id}\otimes\Delta)$$
$$= (\mathrm{id}\otimes\mu)\circ(\mathrm{id}\otimes\mathcal{P})\circ(\mathcal{P}\otimes\mathrm{id})\circ(\mathrm{id}^{\otimes 2}\otimes\mu)\circ(S\otimes\mathrm{id}^{\otimes 2}\otimes S)\circ(\mathrm{id}\otimes\Delta\otimes\mathrm{id})\circ(\mathrm{id}\otimes\Delta)$$
$$= (\mathrm{id}\otimes\mu)\circ(\mathrm{id}\otimes\mathcal{P})\circ(\mathcal{P}\otimes\mathrm{id})\circ(\mathrm{id}^{\otimes 2}\otimes\mu)\circ(S\otimes\mathrm{id}^{\otimes 2}\otimes S)\circ(\mathrm{id}^{\otimes 2}\otimes\Delta)\circ(\mathrm{id}\otimes\Delta)$$
$$= (\mathrm{id}\otimes\mu)\circ(\mathrm{id}\otimes\mathcal{P})\circ(\mathcal{P}\otimes\mathrm{id})\circ(S\otimes\mathrm{id}\otimes(\eta\circ\epsilon))\circ(\mathrm{id}\otimes\Delta)$$
$$= (\mathrm{id}\otimes\mu)\circ(\mathrm{id}\otimes\mathcal{P})\circ(\mathcal{P}\otimes\mathrm{id})\circ(S\otimes\mathrm{id}\otimes 1)$$
$$= \mathcal{P}\circ S_1, \tag{A.28}$$

where we have again used for each equality: anti-homomorphism property of $S$ with respect to the product; $\mathcal{P}\circ S_2=S_1\circ\mathcal{P}$; commutativity of $\mu_{23}$ and $\Delta_1$; $\Delta_1\circ\mathcal{P}_{23}\circ\mathcal{P}_{12}=\mathcal{P}_{34}\circ\mathcal{P}_{23}\circ\mathcal{P}_{12}\circ\Delta_2$;

$\mu_{34} \circ \mathcal{P}_{34} \circ \mathcal{P}_{23} \circ \mathcal{P}_{12} = \mathcal{P}_{23} \circ \mathcal{P}_{12} \circ \mu_{34}$ and commutativity of $S_{13}$ and $\Delta_2$; cocommutativity of the coproduct; compatibility of the antipode; compatibility of the counit with respect to the coproduct and that $\mu(\cdot \otimes 1) = \text{id}(\cdot)$. Plugging this result into eq. (A.27), we see that it evaluates to the last line of eq. (4.23).

Last, we explain how the Fay-like identity for kernels (5.1) follows from the Fay-like identity for generating functions (4.24): We begin by expanding the generating functions in eq. (4.24) in words $(-1)^s b_{i_1} \ldots b_{i_r} \otimes b_{p_s} \ldots b_{p_1}$. This can be done term by term, exploiting the definitions of the component forms and applying the respective Hopf algebra morphisms. The result for the distinct terms is given by

$$
K_{(\mathfrak{b}_1)k}(u,v) K_{(S\mathfrak{b}_2)pp'}(w,x) \Big|_{p'=p}
$$
$$
= \sum_{r,s \geq 0} (-1)^s \omega_{i_1 \cdots i_r i}(u,v)\, \omega_{p_1 \cdots p_s pp'}(w,x) \Big|_{p'=p} b_{i_1} \ldots b_{i_r} \otimes b_{p_s} \ldots b_{p_1}, \tag{A.29a}
$$

$$
K_{(\Delta \mathfrak{b}_{12})jk}(u,v) K_{(S\mathfrak{b}_2)j}(w,x)
$$
$$
= \sum_{r,s \geq 0} \sum_{m=0}^{s} (-1)^m \omega_{(i_1 \cdots i_r \shuffle p_s \cdots p_{m+1})jk}(u,v)\, \omega_{p_1 \cdots p_m j}(w,x)\, b_{i_1} \ldots b_{i_r} \otimes b_{p_s} \ldots b_{p_1}, \tag{A.29b}
$$

$$
K_{(\Delta \mathfrak{b}_{12})j(\mathfrak{b}_1)k}(u,v) K_{(S\mathfrak{b}_2)j}(w,x)
$$
$$
= \sum_{r,s \geq 0} \sum_{l=0}^{r} \sum_{m=0}^{s} (-1)^m \omega_{(i_1 \cdots i_l \shuffle p_s \cdots p_{m+1})j i_{l+1} \cdots i_r k}(u,v)\, \omega_{p_1 \cdots p_m j}(w,x)\, b_{i_1} \ldots b_{i_r} \otimes b_{p_s} \ldots b_{p_1},
$$
$$
\tag{A.29c}
$$

$$
K_{(\Delta \mathfrak{b}_{12})j}(u,v) K_{(S\mathfrak{b}_2)j(\mathfrak{b}_1)k}(w,x)
$$
$$
= \sum_{r,s \geq 0} \sum_{l=0}^{r} \sum_{m=0}^{s} (-1)^m \omega_{(i_1 \cdots i_l \shuffle p_s \cdots p_{m+1})j}(u,v)\, \omega_{p_1 \cdots p_m j i_{l+1} \cdots i_r k}(w,x)\, b_{i_1} \ldots b_{i_r} \otimes b_{p_s} \ldots b_{p_1},
$$
$$
\tag{A.29d}
$$

where the additional sum over $m$ (and $l$) in the second, third and fourth expansion arises due to the different possibilities of distributing the letters to the two generating functions. We can now read off the coefficient of the word $(-1)^s b_{i_1} \ldots b_{i_r} \otimes b_{p_s} \ldots b_{p_1}$ for each distinct term. In particular, substituting the appropriate arguments and expansions for all the terms in eq. (4.24), we directly arrive at eq. (5.1).

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
