# Peer review of "Higher-genus Fay-like identities from meromorphic generating functions"

_SciPost Physics_

## Round 1 · Referee Report · Anonymous (Referee 1) · 2024-12-14

Report

Defining and manipulating polylogarithms on Riemann surfaces in now a timely topic in quantum field theory and string theories. The standard approach is to start from a generating function for the integration kernels. The polylogarithms are then iterated integrals of these integration kernels. Of particular interest are then additional relations between the integration kernels. One may classify these relation by the number of points on the Riemann surface they depend upon, with a dependence on two or three points being the most prominent examples. Furthermore, one may classify these relations by the degree in the integration kernels, e.g. being linear, quadratic, etc. in the integration kernels. The most interesting ones are the quadratic relations depending on three points. These are called "Fay-like" identities and the main topic of this paper. Fay-like identities have been conjectured recently.

In this paper the authors prove for Enriquez meromorphic generating function quadratic relations for component forms (which they define in the paper) and show that these relations comprise all Fay-like identities conjectured previously for this case (i.e. for the Enriquez meromorphic generating function). Furthermore they show that there cannot be any further identities. The last section gives an important application of their result ("removing $z$ from the labels), this is extremely helpful for manipulating polylogarithms on Riemann surfaces.

In summary, this is a very valuable addition to the literature. Before recommending the manuscript for publication I would like the authors to consider the following suggestions, which would render the manuscript more readable:

  1. I have a hard time accepting definition 1 to be a definition. It only says that there exists a one-form $K(z,x)$ and that this one-form is unique, but it does not say how it looks like. This might be acceptable to mathematicians, but for a publication in a physics journal one expects a proper definition.

  2. The authors might want to comment why the only elements of the algebra $\frak{t}$ appearing in eq.(2.3) are the ones with a number of $b$'s on the left and exactly one $a$ on the right.

  3. The authors might want to comment on the meaning of the denominator $1 \otimes b_i-b_i \otimes 1$ in eq.(3.8). A priori one just has the algebra $\frak{b} \otimes \frak{b}$ and no division.

Recommendation

Ask for minor revision

  • validity: -
  • significance: -
  • originality: -
  • clarity: -
  • formatting: -
  • grammar: -

Author:  Konstantin Baune  on 2025-01-15  [id 5124]

(in reply to Report 1 on 2024-12-14)

(see attached response letter)

Attachment:

ResponseLetter_gjIQ58O.pdf

---

## Round 1 · Referee Report · Anonymous (Referee 2) · 2024-12-15

Strengths

  1. Well-written and well-structured
  2. Clear connection with contemporary works in Mathematics and Physics
  3. Several examples and worked out proofs
  4. Paves ground for new work

Weaknesses

  1. Currently (December 15, 2024) the reader cannot readily verify numerically one of the concrete examples: Example 14, Equation (7.17)

Report

This work deals with obtaining "Fay-like identities" for the meromorphic integration kernels of polylogarithms defined on Riemann surfaces of genus h>1. These are generalizations of partial-fraction-identities that rational functions (on the Riemann sphere) satisfy, and also Fay identities that the Kronecker-Eisenstein series satisfy, at genus-one, hence the name. Knowledge of these "Fay-like identities" is crucial for performing computations in quantum field theory and string theory: Feynman integrals and string amplitudes.

The authors find these quadratic identities (and also linear and differential identities), and with solid mathematical precision prove that they have exhausted the space of such quadratic relations. The mathematical proofs are written such that the arguments are transparent and it is clear what the goal of each step of the proof is.

This proves all the conjectured Fay-like relations put forth in [2407.11476], who started with a setup of non-meromorphic but single-valued integration kernels of polylogarithms, and moreover confirms that there are no more such identities.

The authors also make a careful study of their genus-h "Fay-like identities" when h is taken to be equal to 1. In such case, they confirm that they recover the genus-one Fay identities that Kronecker-Eisenstein series.

In a last section before concluding, the authors consider some applications of their Fay-like identities to higher-genus polylogarithms. They pay particular attention to the so-called "z-removal identities," and study them in detail, including giving some concrete examples. One of the most interesting examples is Example 14, which is an example that involves a multiple zeta value, -zeta(2)/(2*pi*i)^2 = 1/24.

Here is a nitpick:
The authors remark that equation (7.17) having this 1/24 can be checked numerically by the methods of [2406.10051]. It is my understanding as of December 15, 2024, that no code accompanies [2406.10051], so it rests on the reader to implement the methods of [2406.10051] themselves. I was not personally able to implement these in a reasonable amount of time. I don't think your average interested reader will not be able to numerically check these polylogarithmic identities by themselves easily.

Overall, I'm very happy with this work, and have enjoyed reading it.

Requested changes

  1. Grammatically, there is the use of "between" when I would expect "among" in several places in the text. I would suggest using "among" instead of "between" in several places.

For example, this sentence in page 3 read better with "among" instead of "between": "The properties of the connection form (including its flatness) will then imply relations between integration kernels, which in turn lead to functional relations between polylogarithms and associated special values."

  1. Equation (7.17) is a concrete outcome of several computations in this work. It would greatly benefit the reader (and myself as a referee) if they could readily check this numerically, so here is my request: Could you include some code to numerically evaluate depth-2 polylogarithms at genus-2, so the readers can check Equation (7.17)? If this is not possible, I would like to know too.

Recommendation

Ask for minor revision

  • validity: high
  • significance: high
  • originality: high
  • clarity: top
  • formatting: perfect
  • grammar: excellent

Author:  Konstantin Baune  on 2025-01-15  [id 5123]

(in reply to Report 2 on 2024-12-15)

(see attached response letter)

Attachment:

ResponseLetter.pdf

---

## Editorial Decision

resubmitted